# Natural Fibre and Hybrid Composite Thin-Walled Structures for Automotive Crashworthiness: A Review

**DOI:** 10.3390/ma17102246

**Published:** 2024-05-10

**Authors:** Monica Capretti, Giulia Del Bianco, Valentina Giammaria, Simonetta Boria

**Affiliations:** Mathematics Division, School of Science and Technology, University of Camerino, Via Madonna delle Carceri 9, 62032 Camerino, Italy; giulia.delbianco@unicam.it (G.D.B.); valentina.giammaria@unicam.it (V.G.)

**Keywords:** energy absorbers, natural fibre composites, hybrids, tubular structures, automotive applications

## Abstract

Natural fibres, valued for their low density, cost-effectiveness, high strength-to-weight ratio, and efficient energy absorption, are increasingly emerging as alternatives to synthetic materials in green composites. Although they cannot fully replace synthetic counterparts, like carbon, in structural applications due to their inferior mechanical performance, combining them through hybridization presents a potential solution. This approach promotes a balance between environmental benefits and mechanical efficiency. Recently, the transportation sector has shifted its focus towards delivering lightweight and crashworthy composite structures to improve vehicle performance, address safety concerns, and minimise environmental impact through the use of eco-friendly materials. The crashworthiness of energy absorbers, typically thin-walled structures, is influenced by several factors, including their material and geometric design. This paper presents a comprehensive overview of recent studies focused on the crashworthiness of fibre-reinforced, thin-walled composites under axial crushing. It explores different aspects, such as their materials, cross-sections, stacking sequences, triggering or filling mechanisms, and the effect of loading rate speed. Emphasis is placed on natural-fibre-based materials, including a comparative analysis of synthetic ones and their hybridization. The primary objective is to review the progress of solutions using green composites as energy absorbers in the automotive industry, considering their lightweight design, crashworthiness, and environmental sustainability.

## 1. Introduction

Recently, composites have been widely used in a variety of industries, including the automotive [1,2,3], aerospace [4,5,6], and civil engineering [7,8,9] sectors, due to their high specific strength, corrosion resistance, energy absorption capability, and flexible design. In the transportation field, in recent decades, there has been a growing interest among researchers and industries in the evolution of passive safety systems [10,11,12,13,14]. Population growth, the expansion of suburban areas, and urbanisation are contributing to an uncontrolled increase in traffic volume in metropolitan areas. This rise inevitably leads to heightened environmental degradation and safety hazards, including a greater likelihood of accidents. More stringent safety measures and traffic regulations are therefore required. In this regard, thanks to the implementation of new laws and strategies, the global status report on road safety 2023 [15] stated that there has been a 5% reduction in the number of global road traffic deaths since 2010. Notably, the European region experienced the most significant decrease in road deaths. Specifically, the 17th Road Safety Performance Index Report [16] attested to a 22% reduction in road accident fatalities between 2012 and 2022. However, such results may not be able to effectively halve this number by 2030, as targeted. Therefore, there is an urgent necessity to develop improved strategies aimed at enhancing road safety and mitigating the impact of collisions. Recent years, indeed, have show heightened interest from researchers and industries in investigating and enhancing advanced passive safety systems, such as crash-boxes, and bumper and pillar reinforcements [17,18,19,20]. To be deemed safe, a vehicle has to demonstrate the ability to dissipate propagated energy during an impact event. The design of passive safety system components, in particular, must prioritise crashworthiness, which is defined as the ability to absorb energy during a crash, thereby minimising the extent of damage and injuries [21,22].

Thin-walled crashworthy components, due to their inherent structure, typically exhibit a progressive deformation of walls, which promotes an efficient energy absorption and a stable response during the impact. Theoretically, when adopting a constant polygonal cross-sectional shape, the greater the number of edges, which can range from triangle to octagon until reaching a circular shape, the greater the energy absorption [23]. While this concept is effectively applicable to metal structures, in the case of composites, the manufacturing challenges and potential defects in complex composite structures, combined with the undesired concentration of stress near sharp edges, make the use of intricate angular geometries inappropriate. Consequently, circular [24,25,26,27,28,29,30,31,32,33,34,35,36,37,38,39,40,41,42,43,44,45,46,47] and square [33,48,49,50,51,52,53,54,55,56,57] sections are the most widely adopted in this field. In fact, the present overview of tubes focuses on two such cross-sectional shapes. Moreover, cones and truncated cones [58,59,60,61,62,63,64] are commonly adopted as energy absorbers. Unlike tubes, they do not require crush initiators, known as triggers, to initiate failure across the point of stress and facilitate progressive crushing. Given the complexity of the energy absorption process, designers of energy-absorbing structures must possess a thorough understanding of of the structure–property relationship in these composite systems. Therefore, it is crucial to discern the primary mechanics involved in the crushing of composites [65,66].

A comprehensive review of the existing literature is provided, focusing on the experimental aspects of the crashworthiness of composite tubes under axial crushing. This study conducts a detailed analysis of the influence of various geometric and material parameters on the energy absorption capabilities of these structures. Furthermore, a comparative assessment is carried out of the energy absorption potential of natural-fibre-reinforced composites (NFCs) in comparison to typical synthetic counterparts, such as carbon-fibre-reinforced polymers (CFRPs) and glass-fibre-reinforced polymers (GFRPs). While synthetic fibres, such as carbon, demonstrate exceptional performance in automotive applications, this study aims to highlight the potential effectiveness of NFCs as energy absorbers. In recent years, extensive research has delved into the feasibility of using natural reinforcements either as complete substitutes for synthetic fibres or in hybrid configurations [67,68,69]. Despite their sustainability, integrating natural fibres into structural applications, instead of synthetic ones, poses significant challenges. Despite the recent trend of using natural fibres in structural components, there has been a lack of extensive exploration into the crashworthiness properties of thin-walled structures or more complex crash-boxes as compared to their synthetic reinforcement counterparts. For these reasons, building upon the available insights gathered from the literature, this work focuses on exploring strategies to enhance the incorporation of bio-based materials in energy absorber structures within the automotive industry.

The paper is organized as follows. Section 2 delves into the application of NFCs in the automotive industry. Following that, a dedicated section defines crashworthiness and explores its influencing factors. Subsequently, Section 4 reports an examination of the literature results concerning the energy absorption performance of tubular (with circle and square sections) and conical composite structures. This investigation encompasses an assessment of the crush load–displacement characteristics of these components, as well as an exploration of the failure modes outlined in Section 5. Throughout this examination, NFCs and hybrids play a pivotal role in the material assessment process, and are continually compared to synthetic alternatives. Section 6 discusses challenges and potential research directions to optimise the performance of NFC energy absorbers. Lastly, the paper concludes in Section 7 by summarizing the key insights, drawing overall conclusions, and discussing future challenges.

## 2. NFCs and Automotive Industry

The inherently lightweight nature of composite materials is a crucial attribute, which is harnessed in the automotive sector for several compelling reasons. In the pursuit of an enhanced vehicle performance, with a reduction in fuel and oil (or electricity for modern engines) consumption and emissions, it is imperative to design each component with the goal of minimising its weight. In this context, composites are increasingly being applied in various vehicle parts to achieve both a reduced weight and optimal vehicle performance, replacing traditional metallic materials, such as steel and aluminium [70]. The superior energy absorption performance of composites, even under high loading rates, is based on a combination of complex crushing mechanisms, i.e., transverse shearing, brittle fracturing, lamina bending, and local buckling, depending on the mechanical properties of the materials and their architecture [65,66,71]. Failure of crushed composite tubular structures typically includes fibre and matrix fracture, fibre–matrix debonding, and delamination, whereas metals mainly exhibit plastic deformation with extensive buckling folding (concertina pattern and diamond modes [72]). For these reasons, and due to their outstanding mechanical properties, composites are increasingly replacing metals, despite their brittleness compared to the ductility of the latter. In particular, in the recent decades, GFRPs and CFRPs have been the most widely used synthetic-fibre-reinforced composites in the production of lightweight crashworthy components [73,74,75,76,77,78].

At present, the drive for high-performance products is coupled with an increasing focus on achieving sustainability goals. Both local and global regulations are progressively requiring the implementation of eco-friendly practices. This shift towards sustainability underscores a wider acknowledgment of the necessity of reducing the environmental impacts across diverse sectors. Industries are now compelled to seek materials and processes that not only deliver a good performance but also have a minimized ecological footprint throughout their lifecycle. Basically, the convergence of performance-oriented needs and sustainability imperatives has spurred a paradigm shift in the materials landscape, pushing industries and researchers to explore innovative, eco-friendly composite materials that meet society’s evolving needs while minimising the environmental impact. In this context, it is important to assess the environmental performance of a product through the life cycle assessment (LCA) technique, which encompasses stages such as raw materials’ extraction, processing, manufacturing, transport, use, re-use, maintenance, and recycling [79,80]. Given the circular economy system’s model of production and consumption and the extended lifespan of composites, particular emphasis should be placed on efficiently reusing the energy still embodied in the materials. Therefore, special attention should be paid to the end-of-life phase of fibre-reinforced polymers (FRPs), as these materials are challenging to recycle. Although mechanical, thermal, or chemical recycling are possible, there are also many limitations associated with these methods [81,82], such as the high energy consumption and costs during the processes and material degradation (i.e., fibre breakage, polymer degradation, and the dispersion of long-fibre reinforcements) [83,84]. Recycled carbon fibres, for instance, when pyrolyzed, generally exhibit a significant loss of strength and stiffness compared to their virgin precursors [85,86]. Other disadvantages of recycling that should be considered pertain to the production, use, and disposal of solvents during chemical recycling processes, aiming to break down composite materials into their constituents. These aspects need to be thoroughly assessed for their potential negative environmental impacts, such as pollution and waste [87,88,89].

However, natural fibres present benefits compared to their synthetic counterparts in terms of their environmental sustainability, reduced environmental impact, energy efficiency, health and safety, and market demand [90]. For these reasons, considering the factors of vehicle performance, safety, renewability, and cost, composites using natural fibres are currently regarded as the most promising alternative solution to synthetic reinforcements in composite applications. Natural fibres, such as flax, hemp, kenaf, cotton, and silk, offer several advantages. First, the extraction of raw materials for NFCs is simpler and more environmentally friendly, and the materials have the added advantage of being biodegradable. Natural fibres can decompose naturally over time through biological processes. Biodegradability can minimise waste, contributing to a circular economy by returning resources to the environment instead of having them accumulate in landfills or oceans. This reduces the environmental burden and promotes a more sustainable approach to material usage. Furthermore, they are more readily available, less expensive than synthetic counterparts, and possess good specific strength and stiffness except when compared to glass or carbon.

Additionally, they exhibit favorable thermal and acoustic insulation properties. Moreover, the production of NFCs can yield good mechanical qualities, and their manufacturing process requires less energy compared to synthetic alternatives [91]. Specifically, these fibres are also carbon-neutral, as there is no net difference between the amount of ${\mathrm{CO}}_{2}$ that is absorbed and produced. Commonly employed thermoplastic polymer matrices for bio-fibres include polyethylene (PE) [92], polypropylene (PP) [93], and polyvinyl chloride (PVC) [94]. Thermosetting matrices include phenolic [95], polyester [96], and epoxy resins [97]. Additionally, natural fibres, being sensitive to environmental conditions like humidity and temperature, frequently need pre-treatments to ensure proper adhesion to matrices [98,99,100]. Nevertheless, it is crucial to note that not all NFCs can be considered fully biodegradable and sustainable. In fact, traditional matrices used in NFCs, such as petroleum-based polymers, may hinder the biodegradability and sustainability of the composite as a whole. By using bio-based polymers as the matrix material, the composite can achieve a higher level of sustainability. Bio-based polymers are derived from renewable resources, such as plants, and have the potential to biodegrade at the end of their life cycle, reducing their environmental impact. Additionally, bio-based polymers typically require less energy to produce compared to their petroleum-based counterparts, further enhancing their eco-friendliness [101]. For these reasons, the increasing emphasis on sustainability in recent decades has led researchers and industries to intensify their efforts to include bio-derived feedstocks into epoxy systems. Despite the challenges associated with this bio-incorporation, existing studies on NFCs applications have shown positive enhancements in the mechanical properties through the addition of natural fibres. This improvement is attributed to the enhanced compatibility of natural fibres and the bio-matrix compared to traditional epoxy composites [102]. However, the adoption of fully bio-based composites in structural applications like the automotive industry is still limited. This could be due to factors such as cost, the availability of suitable materials, and the need for further research and development to optimise their performance and the manufacturing processes for specific applications. Therefore, in the design phase, a comprehensive grasp of various biodegradable polymers and composites, including their properties, production methods, and degradation processes, is essential to effectively implement a Design for a Life (DFL) approach. This understanding is crucial for accurately evaluating and assessing their life cycle and positive impact [103].

For the aforementioned reasons, researchers have recently been dedicated to exploring advanced green composite structures through the incorporation of natural fibres, which are used as energy absorbers in automotive crashworthiness applications, as outlined in the following sections. NFCs have already found application in the production of various vehicle body panels, including door and window frames, columns, ceilings, seat backs, and trunk liners [104,105,106,107,108]. In the automotive industry, bast fibres like flax and hemp, along with jute and kenaf, are the most used. Several prominent car manufacturers have been investing in high-performance composites based on natural fibres [104,109,110]. In 2019, for example, Porsche produced the first global vehicle for motorsports featuring exterior components crafted from hemp and flax natural-fibre-reinforced composites. Audi incorporates flax and sisal mat in its door trim panels. BMW Group incorporates NFCs into its automobiles. Notably, BMW 7 series cars contain renewable raw materials, with flax and sisal replacing fibreglass, in the interior door lining’s panels [111,112,113]. However, the use of purely reinforced natural fibre composites in structural applications (crash-boxes, side impact and pillar structures, etc.) is still not entirely feasible and remains a future challenge due to the poorer mechanical performance of natural reinforcements compared to synthetic ones, as reported in Table 1. Figure 1 shows the first example of a composite front impact structure that is totally reinforced with flax fibres, realised through the collaboration of YCOM and Bcomp. One promising approach to address the limitations associated with integrating natural fibres into structural applications is through hybridisation with synthetic fibres, which could enhance the properties of the final composite while reducing the carbon footprint [114,115,116]. This strategy aims to achieve a trade-off between efficient performance and a reduced environmental impact of the products.

**Table 1 materials-17-02246-t001:** Mechanical properties of some typical natural and non-natural fibres, including carbon, glass, and basalt fibres [117,118,119].

Fibre	Density [g/cm^3^]	Tensile Strength [MPa]	Young’s Modulus [GPa]	Elongation at Break [%]
Cotton	1.5–1.6	200–800	5.5–15.1	2.1–12
Flax	1.3–1.6	340–1600	8.5–40	1.9–12
Hemp	1.1–1.6	285–1735	14.4–70	0.8–4
Jute	1.3–1.5	385–850	25–81	1.1–3.3
Kenaf	0.6–1.5	223–1191	11–60	1.6–4.3
Oil palm frond	0.6–1.2	20–200	2–8	3–16
Ramie	1.4–1.55	200–1000	41–130	1.2–4
Silk	1.3–1.4	500–2000	8.5–30	15–35
Basalt	2.65–2.8	700–1680	70–90	0.5–1.6
Carbon	1.4–1.78	3400–4800	230–425	1.4–1.8
E-glass	2.5–2.55	2000–3500	70–73	0.5–3

**Figure 1 materials-17-02246-f001:**
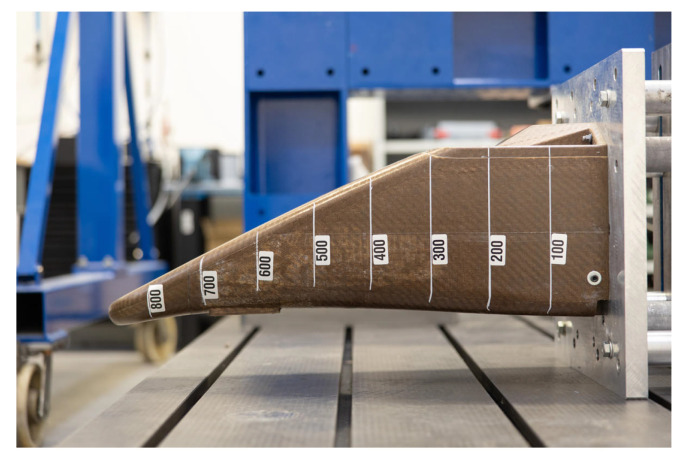
Flax-reinforced composite crash-box for motorsports, using high-performance ampliTex flax fibres [120].

## 3. Crashworthiness

Crashworthiness refers to the ability to absorb energy in a controlled manner during an impact event through failure modes and crushing mechanisms. In the transportation industry, ensuring the crashworthiness of each vehicle component is imperative, as this plays a crucial role in preventing serious injuries or fatalities during accidents. Even with the most advanced active safety systems, accidents cannot be completely prevented for various complex and casual factors. Therefore, there is a demand for systems that can lessen the impact of accidents on occupants. These systems primarily aim to enhance passenger safety by reducing crush and dissipating kinematic energy through controlled deformation processes, primarily converting it into internal energy [121,122,123].

To assess the crashworthiness and ensure the structural integrity of a component, it is essential to conduct crash tests. When looking at the test results, it is important to analyze the load–displacement response to quantify the extrinsic properties of the crushed specimen.

Figure 2 illustrates a typical load–displacement curve, obtained from a tested composite under axial crushing. The crushing event can be delineated into three distinct phases: (1) the pre-crushing zone, (2) the post-crushing zone, and (3) the compaction/failure zone. The pre-crushing zone corresponds to the phase prior to the first failure, which occurs at the first drop of the curve after the peak in the load. The next phase is characterized by the average crushing load. The final phase can be defined by either compaction, which occurs when the tested specimen is completely compressed at the end of the test, as indicated by the final steep positive slope of the curve, or by sample failure, denoted by a negative slope of the curve. In general, traditional passive energy absorbers were typically designed to progressively fail to dissipate energy while maintaining a relatively constant plateau in the load. The deformation process can typically be divided into three stages: elastic, plateau, and densification. During the initial elastic stage, the material deforms elastically in response to the applied load. Following the initial peak load, which signifies the onset of plastic deformation or a sudden increase in load before significant deformation occurs, the progressive collapse or buckling of structural elements within the material characterizes the plateau stage. The densification stage follows the plateau stage and is marked by a rapid increase in load or stress, accompanied by significant deformation and compaction of the material.

The load–displacement curve allows for the direct derivation of the following parameters, which are essential for evaluating the crashworthiness of the composite [124,125,126].

The maximum load recorded during impact (P_max_), excluding the post-crushing zone and the potential increase in the load due to compaction.The total absorbed crushing energy (AE) is the amount of energy abosorbed during the impact event. In dynamic crushing scenarios, this denotes the conversion of all kinetic energy into absorbed internal energy:
(1)$$\mathrm{AE}=\int_{0}^{x_{\max}}P\;\mathrm{dx}$$
where P is the crushing load and $x_{\max}$ is the maximum displacement. Thus, AE represents the area under the load–displacement curve.The specific absorbed energy (SEA) is the AE normalized by the mass of the crushed specimen (m). Thus:
(2)$$\mathrm{SEA}=\frac{\mathrm{AE}}{m}$$The average crush load (P_avg_) is the AE normalized by $x_{\mathrm{post}}$, where $x_{\mathrm{post}}=x_{2}-x_{1}$ is the displacement related to the post-crushing zone (see Figure 2).The crush force efficiency:
(3)$$\mathrm{CFE}=\frac{P_{\mathrm{avg}}}{P_{\max}}$$
quantifies the deviation of the average load from the peak value.

The assessment and comparison of energy absorption ability in composite components primarily relies on SEA, which is an essential measure for prioritising lightweight materials. In general, the higher the SEA, the greater the performance of the component as an energy absorber. This measure is directly proportional to the AE, which quantifies the amount of impact-absorbed energy, and inversely proportional to the mass of the crushed component. Moreover, since AE corresponds to the area under the load–displacement curve, it is directly influenced by the P_avg_ value. Therefore, it is desirable to obtain the highest average load to achieve the maximum absorbed energy. However, a higher P_max_ corresponds to a faster deceleration of the structure during a crash event. To effectively minimise the injuries sustained by a vehicle passenger in a collision, it is crucial to keep this value as small as possible. Consequently, achieving a CFE value close to one indicates a minimal deviation of P_avg_ from the peak. This could help to mitigate the deceleration perceived by passengers.

Therefore, designing both lightweight and crashworthiness involves two interdependent objectives: maximising the energy absorption ability, and minimising the total weight of the structure. To achieve the highest SEA value while limiting the peak load by controlling the CFE, it is important to understand how the geometrical and material characteristics of a composite structure can impact these outcomes. When aiming to maximise the crashworthiness performance of a composite structure, it is possible to optimise the aforementioned parameters, for instance, through a material or topology optimization [127,128,129]. Specifically, by focusing on tubular composite structures, one can address the following properties to achieve an optimal crashworthiness performance [130,131,132]:Geometry;Fibre and matrix material;The presence of a trigger and filling foam;Stacking sequence;Crushing speed.

In the next section, the influence of the above factors on crashworthiness parameters will be discussed in detail. This analysis will be systematically accompanied by an exploration of the advantages and challenges associated with the use of NFC and hybrid configurations, comparing them to some examples of common synthetic-fibre-based composites.

## 4. Thin-Walled Structures under Axial Crushing

Thin-walled structures in a vehicle serve as attenuators, and are typically designed and strategically placed in specific zones to absorb the impact energy and protect the passengers. The analysis below delves into subdividing tubular structures based on cross-sections (including circles and squares), as well as cones and truncated cones. Furthermore, it explores the effects of fibre and matrix properties, triggering mechanisms, foam filling, and loading rates on crushing behavior.

Many studies have investigated the energy absorption of FRP tubes, encompassing both circular and square cross-sections. However, the majority of these studies predominantly utilised synthetic fibres. In recent years, the increased attention paid to NFCs has spurred research in this area, which is the focus of the analyses below. Overall, the results indicate that the geometric shape significantly influences the energy absorption capacity of composite structures. Hereafter, only the axial crushing behavior of composite tubes will be discussed, excluding consideration of lateral crushing, as the focus will be on the higher load-carrying and energy absorption capabilities that are evidenced in the longitudinal case [53,133,134].

### 4.1. Geometry

#### 4.1.1. Circular Cross-Section

Circular cross-section tubes exhibit a better energy absorption ability than square or rectangular cross-sectional ones. Within the stable collapse region, they are more stable than polygonal wall sections [34,57].

When discussing the tube geometry (see Figure 3), it is useful to characterize it according to its inner diameter (D) or its outer diameter. Other important parameters include the thickness (t) of the tube and the number of layers (N) in the laminate. Moreover, other geometric factors to consider are the tube length (L), and the diameter-to-thickness ratio (D/t) or the diameter-to-layers ratio (D/N). These variables collectively influence the energy absorption capabilities of the structure.

Among the existing studies in the literature regarding tubes of natural fibres, it is worth mentioning some works focusing on the use of flax fibre, which exhibits a good crashworthiness performance thanks to its high specific energy absorption, natural damping properties, flexibility, and toughness. In [24,25,26] Yan et al. experimentally investigated the crashworthiness performance of flax/epoxy (F/E) circular tubes. The study emphasises the potential of such structures as energy absorbers, demonstrating promising SEA values when subjected to axial crushing. Various configurations were tested under quasi-static conditions to assess the impact of geometrical parameters on crushing behavior. The best energy absorber exhibited an SEA value of 41 J/g and a CFE of 0.8, with smallest D value among all tested samples and the highest number of layers, as well as the highest R (=L/D) ratio.

These values are greater than those of traditional metallic components and comparable to GFRPs and CFRP, as reported in the literature [42,43,71]. It can be observed that a greater N and a larger L determined an increased energy absorption capability. It also emerged that, with a fixed N, the smaller the value of D, the higher the SEA, as depicted in Figure 4. Furthermore, increasing the length and thickness of the tube resulted in higher values of both P_max_ and CFE. This should be attributed to the greater dissipation of energy over a longer distance and the heightened resistance experienced during the crushing process, especially in cases with a higher number of layers. When D remained constant, a higher value of N resulted in elevated values of P_avg_, AE, and SEA. Notably, the most efficient energy absorption was associated with specimens with the lowest D/N ratio, which held true regardless of the reinforcement used, as reported in [30,39,40,41].

In a later study [27], the same authors also investigated glass/epoxy (G/E) and basalt/epoxy (B/E) tubes in order to compare their performance with those obtained using flax. The influence that the same geometrical parameters has on crashworthiness was observed independently of the type of reinforcement. However, notably, tubes made of flax exhibited a higher SEA compared to those made of basalt, comparable to those made of glass.

Sivagurunathan et al. [28] tested plain weave jute/epoxy (J/E)-triggered circular tubes under quasi-static compression. In addition to the effect of different trigger mechanisms (discussed in Section 4.2), it is interesting to conduct a comparative analysis of the performance observed in this study and the performance of the aforementioned flax case. Flax fibres are stronger than jute ones (see Table 1), so that J/E exhibits a lower average impact load compared to F/E. Consequently, it is reasonable to expect a reduction in crashworthiness performance. Nevertheless, the obtained SEA value was comparable to that observed in the flax case, albeit with a greater D/t ratio. This discrepancy can be attributed to a combination of factors. The larger D/t value in the case of flax, based on prior geometric considerations, typically results in a decrease in crashworthiness performance. However, this effect could be mitigated by the superior crashworthiness efficiency associated with the presence of flax fibres rather than jute, given their typically higher strength. In addition, the influence of the potentially different fibre volume fraction (V_f_) considered in the various studies should also be taken into account. Unfortunately, this information is omitted by the authors; hence, further investigations would be needed to explore this aspect.

Wȩcławski et al. [30] evaluated the effect of different winding angles on the hemp/epoxy (H/E) cylinders subjected to axial crushing. The authors reported that the sample with the lowest angle exhibited the highest average compressive load value; hence, it was found to be the best energy absorber. Nonetheless, no comparable levels of SEA were reached with respect to the other NFCs cases that were discussed.

The failure mechanism of the tested samples, and thus their crashworthiness, remained influenced by the D/t ratio. Notably, a progressive crushing was observed until a ratio of 25, beyond which, as is typical of short and thick tubes, a sudden collapse occurred. From the analyzed results, the use of flax fibre in circular tubes demonstrated effectiveness in terms of crushing behavior, particularly in terms of SEA and crush force efficiency (CFE), when compared to other analyzed natural fibres, such as jute and kenaf, and also glass fibres. In spite of this, carbon fibre significantly outperformed all purely natural-fibre-reinforced composites. In fact, Liu et al. [31] investigated the behavior of carbon/epoxy (C/E) under either quasi-static and dynamic axial crushing. Apart from the differences that emerged due to the rate speed effect (refer to Section 4.4), by halving the D/t ratio from 20 to 10, the SEA value increased from 59 J/g to 74 J/g, under quasi-static loading conditions. The decline in SEA value when using carbon fibre compared to the analyzed natural fibres, particularly the highest-performing flax fibre, despite differences in geometrical characteristics, favors the use of carbon fibre (lower D/t ratio), and amounts to about 55–64%. Conversely, in comparison to glass fibre, flax achieved a slightly superior SEA value (27 J/g vs. 25 J/g).

In this regard, the hybridisation strategy enables the incorporation of eco-friendly constituents, while simultaneously preserving the mechanical performance of the composite. In this context, Attia et al. [29] analyzed the effects of fibre hybridisation and a reinforcement sequence of six layers, with alternating carbon (C), jute (J) and glass (G) reinforcement, in epoxy cylindrical composite tubes tested as energy absorbers. Using jute fibre alone in energy-absorbing composite tubes is ineffective; in fact, the tubes exhibited worse crashworthiness parameters with respect to hybrids or pure glass-based composites. By replacing two layers of jute fibre with one layer of glass and carbon fibre, i.e., J-G-C, the best energy absorber was obtained among all the proposed hybrid configurations. It is worth noting that the hybridisation achieved using J-G-C was better than that of the non-hybrid composite, which was reinforced only with glass. On the other hand, the inverse stacking sequence (C-G-J) resulted in an SEA that was almost 22% lower, despite having the same geometrical parameters. The order of material in ply overlapping, indeed, significantly affects the crashworthiness performance of composite tubes. Supian et al. [32] explored the influence of winding angles on the crashworthiness of kenaf and E-glass embedded in epoxy resin (K-G/E) tubes tested through quasi-static compression. The highest angle yielded the best hybrid energy absorber in terms of SEA and CFE. The beneficial effect of integrating natural fibres is generally expressed by the improvement in impact properties [135,136,137]. In comparison to non-hybrid composite tubes made solely with glass and containing the same number of layers, the incorporation of natural fibres in the K-G/E composite significantly improved its energy absorption capabilities under quasi-static axial crushing. Nevertheless, it should be emphasised that the use of chopped glass fibres as opposed to long filaments may potentially result in an additional reduction in crashworthiness performance. On the other hand, when a single, wet glass, long-filaments band was wound to create a G/E composite, equipped with three additional layers with respect to the previous case, the K-G composite was still superior. With respect to the hybrid case, a slight decrease in the D/t value was observed. However, this was not sufficient to enhance the absorbed energy, as the effect of glass, in fact, lies primarily in its stiffening effect on the structure. In particular, in the case of purely synthetic reinforcement, a higher loading level was observed, accompanied by increased fluctuations, as opposed to the stable crushing load ensured by the use of hybrids, leading to a lower CFE value.

Figure 5 presents a comparison of the energy absorption capability of flax/epoxy, jute/epoxy, kenaf/epoxy, glass/epoxy, carbon/epoxy, and hybrid/epoxy tubes across various D/t ratios. In this regard, when comparing the SEA values from the described studies, it is crucial to ensure they were evaluated at the same or a comparable D/t value. This is important as the energy absorption capability is significantly influenced by the geometric parameters of the tubes. Crushing performance can be also adjusted as desired by manipulating these parameters as well as the material choice. Moreover, all P_avg_ values used to assess CFE are based on the same crushing stage, specifically the post-crushing phase. According to the load–displacement response reported by the authors, these values are subsequently re-estimated if they do not pertain to the aforementioned crushing window, as was achieved for [32].

In all the previously analyzed results pertaining to composites with epoxy resin as the matrix, thermoplastic alternatives have also been explored. Lopèz-Alba et al. [39] described the performances of circular cross-sectional tubes under the quasi-static and dynamic impact of woven flax fibres, as well as a non-woven mat of hemp and kenaf fibres impregnated via the compression molding of High-Density Polyethylene (HDPE) or the Polylactic Acid (PLA) matrix. The different resins’ mechanical properties (see Table 2) meant that they were not equally effective as a thermoplastic matrix for composite tubes under axial impact.

Indeed, in both static and dynamic tests, HDPE-based materials showed a much smaller profile stiffness and crushing load than PLA-based ones. This suggests that the stiffness of the chosen matrix is crucial to ensure stable composite failure and a good level of absorbed energy. In this sense, the HDPE stiffness is too low to guarantee stability during impact. In particular, six or four layers of woven flax/HDPE exhibit an SEA value of only 9 J/g under quasi-static compression, while the use of PLA for the same structure guarantees an increase until 29 J/g.

Finally, a radial corrugation of tubes has been found to be able to enhance the energy absorption capability and the stability at the end of the crushing process under the same conditions regarding the number of layers and length [37,38].

#### 4.1.2. Square Cross-Section

While the majority of studies focusing on the crashworthiness of elementary thin-walled composite structures primarily examined circular tubes, there is merit to examining square cross-sections as well. This is due to their advantages in terms of their ease of assembly with other components and simplified manufacturing process.

A square tube can be geometrically characterized by the length (L), the square width (L_1_), the inner radius (r), and the thickness or number of layers, as outlined in Figure 6.

Eshkoor et al. [48,49] studied the crashworthiness performance of silk/epoxy (S/E) square tubes. The inferior mechanical properties exhibited by silk fibre in comparison to previously analyzed natural and synthetic fibres (as outlined in Table 1) cast doubt on its potential to achieve a promising crashworthiness performance. Notably, even in the most optimised configuration, silk fibre yields an SEA value of only 5 J/g and a CFE of only 0.4. Woven ramie/epoxy (R/E) tubes of three different lengths (50 mm, 80 mm, and 120 mm) were tested under quasi-static crushing in [50]. The crashworthiness analysis revealed that the shortest tube exhibited the highest SEA value of 5 J/g, comparable with the previous S/E case. However, overall, the length did not significantly affect this energy value, as the two other, longer tubes showed values that were only 6–8% smaller. Laban and Mahdi [53] explored the performance of cotton fabric fibres embedded in epoxy resin (Ct/E) in rectangular tubes with varying aspect ratios (a) between two dimensions, selected at random, between 0.5 and 2.6. Although the highest P_avg_ was attained for a = 2, the lighter square case proved to be superior from the perspective of CFE and SEA, which were approximately half those obtained in the previous ramie case. Sivagurunathan et al. [51], after having examined circular tubes, as explained above, then analyzed the effect of a square cross-section on J/E with the same trigger mechanisms. In this case, no significant decrease in crashworthiness was observed with respect to cylinders. Albahash et al. [56] tested hybrid J-G-reinforced epoxy by varying the width of the square cross-section from 50 mm to 100 mm while maintaining the same thickness. This resulted in the $L_{1}$/t ratio being halved and reduced the SEA value by almost 50%.

To assess the varied performance characteristics provided by carbon fibre, it is pertinent to reference some exemplary studies. Mamalis et al. [34,35] and Yang et al. [36] compared the effect of cross-section geometry on composite tubes, enhancing the superiority of circle with respect to the square section in terms of SEA. Regarding maximum load, the square tubes presented a higher value (97 kN) than the cross-sectional one (48 kN). On the other hand, in contrast to the circular tube, the load of the square ones exhibited a sudden reduction shortly after the peak, followed by a slow and steady saturation before progressive crushing. The circular tubes, instead, presented better stability, which resulted in a better crashworthiness performance than the square tubes, especially in terms of SEA and CFE. The computed SEA values were 74 J/g and 55 J/g for circular and square tubes, respectively, while CFE showed a decrease of approximately 30%. Palanivelu et al. [33] reported the differences in the behavior of square and circular glass/polyester (G/P) tubes following quasi-static crushing. Circular tubes showed controlled progressive failure modes compared to the less stable pattern of square tubes, which have a lower energy absorption capability. In particular, SEA underwent a decrease of 60%.

Liu et al. [54] assessed the crushing response of C/E square tubes. The tested structures exhibited a significantly higher energy absorption capability with respect to the natural fibres discussed above, with an increment in SEA value of 60% with respect to the best J/E case. The SEA and CFE values of all examined fibre-reinforced epoxy square tubes are collected in Table 3 in terms of material, L, and width-to-thickness ratio (L_1_/t).

Mache et al. [55] experimentally studied the axial quasi-static impact of square and double-hat-shaped sections of jute/polyester (J/P) and G/P components. When using the same material and different shapes, the tested samples reached comparable levels of SEA around 20 J/g. It is interesting to note that jute tubes generated more P_max_ than glass ones, while the latter presented a larger cross-sectional area. Consequently, overall, they reached comparable P_avg_ values.

Generally, it emerged from the literature that circular cross-sections exhibit a superior energy absorption performance in thin-walled structures compared to square cross-sections. This arises from the geometric irregularities inherent in the square shape, which lead to higher concentrations of stress at its corners. A circle, as the ultimate form of a polygon, offers the most optimal stress distribution due to its continuous profile. When designing a more intricate crash absorber structure, it is advisable to mitigate sharp edges as much as possible by incorporating curved geometries in other areas, as well as in the cross-section itself. Overall, the effect of L_1_/t on energy absorption performance reflects what is observed in the case where D/t is used in circular tubes.

#### 4.1.3. Cones and Truncated Cones

Conical structures are more suited to crashworthiness applications than tubes since they do not need crush initiators. Indeed, for such structures, the crush starts from the the vertex of the cone, where the highest stresses occur [63]. They allow for the progressive dissipation of energy due to their inherent gradual tapering, resulting in controlled deformation upon impact.

A schematic view of cone and truncated cone geometries is presented in Figure 7. The semi-vertex angle (α) has a significant influence on the crashworthiness performance of the structure, as well as the height (h), the thickness (t), the lower internal diameter (D_2_), and the upper internal diameter (D_1_) for truncated conical components.

Mahdi et al. [58] investigated the effect of the vertex angle ($\alpha =0^{\circ },\;6^{\circ },\;12^{\circ },\;18^{\circ }$) on the quasi-static crushing behavior of oil palm frond fibre/epoxy (OP/E), G/E, and C/E truncated cones when axially compressed. The crushing behavior of conical structures causes them to be very sensitive to variations in the cone vertex angle. The variables h and $D_{2}$ were fixed to be equal to 110 and 100 mm, respectively. The greater the cone angle, the smaller the SEA and P_max_ values, while the P_avg_ increases. The experimental results revealed the superiority of cones with the smallest vertex angle, i.e., the cylinder. In particular, OP/E cones with the highest angle fell short of circular tubes made of the same material in terms of SEA by about 50%. By analysing the load–displacement response, greater fluctuations were observed when using cones, which became more intense as α increased; hence, a greater stability was ensured with the use of cylinders. Clearly, the significantly lower modulus of the oil palm frond reinforcement, in comparison to glass and carbon, is reflected in the reduced crashworthiness performance, with a SEA value of only 3–7 J/g, which is 77% lower than that of the C/E composite. Subsequently, the authors [59] investigated the effect of palm oil and also coconut coir fibres on cones, here embedded in polyester resin (OP/P and Cc/P, respectively), with α varying from 0° to 60°. The increase in α generated a decrease in CFE. OP/P cones exhibited greater CFE and greater resistance to crushing loading, resulting in an improved energy performance. This was attributed to its mechanical properties, failure mechanism, and structural integrity. On the other hand, the SEA values obtained for both materials, due to the reinforcement and matrix type, were below unity, making them unsuitable for effective energy absorption applications.

Khalid et al. [61] explored the energy absorption performance of Ct/E and G/E cones, varying the angle α to 5°, 10°, and 20°. Large load oscillations were observed for both materials. However, a much lower performance was obtained in the case of Ct/E compared to G/E due to its inferior mechanical properties. Ismail and Othman [62] examined the experimental quasi-static crushing of coconut coir-reinforced polyester (Cc/P) truncated cones, varying the α angle from 5° to 20°, and varying the V_f_ from 20% to 40%. The higher V_f_ enhanced the energy absorption of composites, leading to a SEA value of 9 J/g for V_f_ = 40%. Meredith et al. [60] experimentally tested dynamic axial crushing NFCs cones, directly showing the significant V_f_ influence on the energy absorption capability of F/E, H/E, and J/E cones. Consistent with the previous observations, the unwoven H/E samples with the greatest V_f_ (47%) exhibited the highest SEA value (54 J/g). It is worth observing this obtained value of absorbed energy was comparable with that obtained by carbon reinforcements (56 J/g). However the AE trend was not similar for the H/E specimens, as V_f_ increased then AE decreased. The different effects of dynamic loading conditions compared to quasi-static conditions on crashworthiness will be provided below (Section 4.4).

From the synthetic fibres perspective, Ochelski et al. [63], using quasi-static compression to test C/E and G/E truncated cones, found that as the cone vertex angle increased, the SEA value dropped because the bending momentum of the layers increased. Hence, the best impact-energy absorber is still the cylinder (α = 0°). On the other hand, the P_avg_ value still showed an incremental increase with the cone angle. Regarding the previously discussed geometries, the greater the thickness, the higher the bending stiffness, meaning that there was also a significant increase in the crushing load, the P_avg_, and the P_max_ values, which, in turn, positively affected the SEA value. Finally, due to its superior mechanical properties, carbon fibre guaranteed the best result in terms of energy absorption performance. In this case, the SEA value is higher, by about 20%, than that of G/E.

For the hybrid material configuration, it worth mentioning the study provided by Israr et al. [64] on flax-glass/polyester (F-G/P). A SEA value of 20 J/g and a CFE of 0.7 were obtained for cones with α = 30° and an intercalation stacking sequence of flax and glass ([F/G/F/$G_{2}$/F/G/F]), which outperformed two others where flax reinforcements composed the outermost layers or core. Table 4 summarizes the different energy absorption capabilities, under quasi-static loading conditions, of the analyzed NFCs and C/E, G/E cones.

### 4.2. Trigger and Foam

#### 4.2.1. Trigger

An effective approach to mitigate the high peak load in tubular structures undergoing crushing is to implement a trigger mechanism. This is capable of concentrating stresses in a precise known zone of the specimen (in the case of tubes, usually at one end). In this manner, the failure starts from the point of stress and propagates to all the other parts of the structure. There are two general procedures of realising trigger mechanisms (shown in Figure 8), as follows:A material-removal method, which generates a weakness zone. The main triggers used are chamfer (internal or external; single or double) and tulip with a chosen angle;The insertion of external stiff components, such as plug-type (outward and inward folding caps) triggers.

Generally, the trigger type and angle β for a fixed type (refer to Figure 8) affect the energy absorption of tubes with a box section; for instance, a circle or square section. Ghasemnejad and de la Cuesta [138] analyzed the difference in the crashworthiness performance of a double 60° and 30° chamfer on CFRP circular tubes. As β increased, so did the P_max_, but a slight increase in SEA was observed for a smaller value of β. An increase was also observed from the perspective of CFE. Nevertheless, a 60° tulip trigger achieved a better performance in terms of the crushing loading stability, although it showed an inferior energy absorption performance. The authors also proposed an optimized configuration through preliminary numerical exploration: a 45° bevel. This choice aimed to maximise the benefits of both the increased β and the number of tip angles as much as possible. Sigalas et al. [139] tested G/E circular tubes by varying the chamfer angle β from 10° to 90°. The smaller the β, the better the response, creating an efficient energy absorber with a reduced load drop. Yan et al. [24,27] examined the effect of a trigger system on F/E tubes under quasi-static axial impact. Designing a 45° external chamfer at the upper end of the composite tube made it possible to control the initiation of micro-fracture at the triggered region and, eventually, a stable crush zone can be generated. As shown in Figure 9, the chamfer led to a reduction in P_max_, but no significant improvement in P_avg_ value was observed. In general, a considerable increase in AE and SEA should not be expected when tubes are equipped with a trigger mechanism. Instead, the purpose is to reduce the high levels of deceleration that occur during an impact event by decreasing the value of P_max_. As a result, the variation in force resulting from the P_avg_ value was decreased, leading to a more stable progressive failure. This led to a CFE increase, especially when D was small. For the tested tubes, =the addition of a trigger meant that the tube with minimum D/t was the best in terms of energy absorption (D = 64 mm, N = 6), with only a slight increase in SEA value with respect to the corresponding non-triggered configuration, while showing an improved CFE of 43%.

In [48], the different effects of a plug and four rectangular metallic pieces, used as triggers on square silk/epoxy (S/E) tubes, were investigated. The metal pieces trigger caused a significant reduction in peak load value (from 14 kN to 9 kN), and as expected, there was no remarkable change in SEA. On the other hand, the plug trigger lowered the P_max_ more; however, unfortunately, the SEA decreased.

During the practical implementation of a triggering mechanism within energy absorber structures, a crucial consideration is how to strike a balance between the effectiveness of the peak load reduction, the energy absorption capabilities, and the ease and efficiency of the manufacturing process. For instance, transitioning from a single to a double chamfer may result in a further decrease in the P_max_ value, but may not yield a significant enhancement in energy absorption performance. Conversely, fabricating a steeple could entail additional manufacturing challenges compared to a simple bevel. Therefore, it becomes imperative to assess the intricacies involved in the manufacturing process related to trigger production. This evaluation assumes heightened importance when dealing with natural-fibre-reinforced composites, where production is often hampered by inherent complexities. In that case, a bevel might be preferred to a steeple or other, more complex configurations, being sufficient to obtain a controlled maximum load value, maintaining the energy absorption capability of the composite structure with limited challenges in manufacturing.

Farley [140] stated that C/E circular tubes’ crashworthiness performance could be better enhanced by a chamfer trigger than by a tulip. In [141], a 45° chamfer was compared with a version of the tulip trigger for both circular and square tubes. The results indicated that the chamfer had a greater effect on energy absorption capability in circular cross-section tubes, whereas the tulip trigger exhibited a better performance in square cross-section tubes. A tulip configuration could lead to less stiff laminates than a chamfer or steeple. In cases with a square cross-section, failure might propagate at the sharp edges of the tube, delaying circumferential delamination propagation along the tube. In that case, the tulip is easily fractured under axial crushing, which can foster the propagation of damage throughout the tube length. However, Sivagurunathan et al. [28,51] argued in favor of the use of a tulip trigger over a single or double chamfer when aiming to augment the energy absorption capabilities of either circular or square J/E tubes. While all triggers facilitate progressive failure, the tulip trigger specifically enhanced lamina bending, axial cracks, and fibre delamination, qualities that are typically sought after in a tubular structure, aiming to maximise energy absorption and slow down the crushing process. Also, Palanivelu et al. [141] compared the effect of a tulip or chamfer trigger on circular and square glass–polyester and glass–vinylester tubes. In circular tubes, the chamfer trigger could help to generate uniform circumferential delamination, followed by the axial cracks, which, in turn, led to a greater energy absorption value (more 7–9% of SEA value) compared to than the tulip trigger. On the other hand, when using the tulip trigger, the SEA value increased 17% more than the bevel in square tubes.

G/P circular tubes were analyzed by Jimenez et al. [142], highlighting the effect of different triggers (bevel and tulip) and angles on each type. Only the chamfer trigger showed sensitivity to angle variation, affecting the crashworthiness performance of circular tubes. The best energy absorber corresponded to the 60° bevel trigger, with an increase in SEA value of about 25% compared to 30° and 45° bevels. Furthermore, this configuration also exhibited the highest recorded P_max_ value. Unlike the material type study described above, here, the 60° tulip performed better than all chamfer angles in terms of SEA.

Based on the conclusions reported in the literature, it can be deduced that the preference of one trigger type over another hinges on factors such as component geometry (for example, cross-sectional shape) and the materials involved. Consequently, these factors hold significant sway over the failure mode, which can be effectively reinforced through a triggering mechanism. However, the reported findings in the literature do not offer a systematic means of categorising and prioritising various trigger mechanisms. Further, when working with composite materials, particularly those incorporating natural fibres, additional manufacturing challenges arise compared to when working with synthetic fibres and traditional metallic materials. Considerations of possible voids, fibre quality, and impregnation play a pivotal role in influencing and enhancing the breaking behavior. Therefore, it is crucial to factor in these aspects when analysing failure modes in conjunction with their potential trigger effects. Studies based on the same material type could come to different conclusions [141,142].

#### 4.2.2. Filling Foam

A trigger mechanism is not the only solution to reduce the P_max_ value and enhance the safety of objects experiencing an impact. Thin-walled structures might be filled with a foam filler to obtain a similar result. The effect is significant as long as the foam density is not too low, ensuring a remarkable increase in energy absorption efficiency that is not excessively large to prevent the transition to inefficient failure mode. Once such filled tubes are compressed, the foam-filler expands laterally and generates pressure on the inner surfaces, which contributes to their resistance to the compression loading. Consequently, the foam filler also achieves an increase in the P_avg_ value. Thus, the CFE values considerably increase due to the foam filler, with a simultaneous peak load reduction and increase in average crush load value. The use of a filling is a clever way to improve the resistance of hollow tubes, without significantly increasing the total weight of the component. Combining the foam and trigger effects can increase the crashworthiness performance. Yan et al. [24,27] used polyurethane (PU) foam to fill cylindrical F/E tubes and discovered that they absorb more energy than empty ones. In particular, when paying attention to lightness, the addition of a foam filler also increased the SEA, since, compared to the increment in the mass, much more energy was absorbed due to the foam. It was found that using this together with the foam filler led to a higher P_avg_ than the when using either the triggering or the foam filler alone. Hence, a combination of triggering and filling can obtain a better crashworthiness performance in terms of SEA and CFE than empty tubes. In the best tested flax/epoxy tube, this combination increased P_avg_ from 47.1 kN to 58.2 kN. Additionally, CFE rose from 0.51 to 0.86, and SEA increased from 25.5 J/g to 28.8 J/g. In particular, the addition of a PU filler slightly increases the SEA value of tubes for all tested samples by 5–13%. However, it might be possible that the use of a foam filler alone leads to either better or worse results than the use of foam-triggered tubes. This could be motivated by the fact the trigger does not affect the sustained crush load and the amount of absorbed energy, only the peak load value. The obtained SEA values of PU foam-filled natural F/E circular tubes were larger than those of steel and aluminium tubes with different foam fillers. Moreover, the energy absorption capacities of PU foam-filled natural F/E circular tubes were close to those of G/E, G/P, and C/E circular tubes with the same or different foam fillers, as reported in Table 5.

Wei et al. [143] tested bamboo/epoxy (Bb/E) tubes filled with PU. The loading capacity and structural integrity of the PU foam-filled bamboo tubes were enhanced with respect to the hollow counterpart. However, the load–displacement response was not affected by a substantial variation following the incorporation of foam. From the energy absorption performance perspective, no improvement was recorded; instead, some negative effects were observed, such as a reduction in the ultimate displacement and hindered load sustainment. Specifically, while the foam filler contributed to the enhanced crushing resistance, it also restricted the inward shrinkage deformation of the tubes. Consequently, the splitting failure of composite tubes occurred after reaching the ultimate load, leading to a rapid drop in load and consequently decreasing both SEA and CFE.

### 4.3. Stacking Sequence

Most of the composites used in engineering applications are laminates. By using different layers (or plies) of combined reinforcements, the matrix can be arranged in various orientations with respect to the axis of the composite.Each ply can have different materials and orientations with respect to the composite axes, as outlined, for instance, in Figure 10. In this sense, either unidirectional and fabric plies can be used. Hand lay-up [144,145], resin transfer and infusion molding [146,147], filament winding [30,32,148], and pultrusion [141,149] are among the most common methods for manufacturing composite laminated tubes. While the hand lay-up method is cost-effective, it may result in the formation of voids. Vacuum bag molding can address this concern, although it is not ideal for high-volume production. Vacuum-Assisted Resin Transfer Molding (VARTM) and Vacuum-Assisted Resin Infusion Molding (VARIM) enable high fibre volume fractions but entail high tooling costs. Filament winding is suitable for cylindrical shapes, but handling low fibre angles (from 0° to 15°) is difficult. Pultrusion is cost-effective but yields lower fibre volume fractions than prepregs, and is restricted to constant cross-sections and zero-degree fibre angles. It is not an effective procedure for producing intricate geometries. On the other hand, 3D printing technology offers lightweight potential, customised structural parts, ease in producing complex structures, a reduction in material waste, and relatively low production costs. However, the printing techniques used in additive manufacturing have the drawback of yielding parts with a reduced mechanical strength and crushing performance compared to those produced with subtractive manufacturing methods [150]. Moreover, innovative improved methods, like 4D printing technology, were recently proposed to create smart adaptive structures, which are characterized by their reusability, minimal maintenance, and high energy absorption capacity [151,152,153]. Therefore, each method has its own advantages and limitations depending on its cost, complexity, and end-use. Based on the design of the structure, the consideration of potential techniques should prioritise considerations such as shape complexity, material properties, cost, and production volumes to determine the most suitable solution [154].

Laminated structures exhibit a high in-plane specific strength and stiffness. On the other hand, the mechanical properties could be significantly affected by the delamination phenomenon, which is one of the main crucial failures in crushing composite laminates, together with micro-cracking in the matrix material, fibre–matrix debonding, and fibre fracture. The stacking sequence of laminates precisely defines the order of the ply stacking and orientation. Rearranging plies by the switching angles of fibres creates laminates with different configurations, and thus individual mechanical properties and energy absorption capabilities. The plies sequence can influence the P_max_ and P_avg_ values of the composite structure, and therefore the crashworthiness performance.

Hull et al. [124] demonstrated that C/E cylinders exhibited to a rapid growth in longitudinal cracks in laminate orientations, for example, of +45°/−45° and +55°/−55°, with respect to 0°/90°. The latter orientation showed a higher energy absorption capability. Ochelsky et al. [63] reported that, among the tested configurations of C/E and G/E tubes ([0°_*n*_], [90°_*n*_], [(0°/90°)_*T*_]_*n*_, [90°/0_*n*_/90°], [(0°/90°)_*T*_/0°_*n*_/(0°/90°)_*T*_], [(± 45°)_*T*_/(0°/90°)_*T*_/(±45°)_*T*_]), the best configuration was given by [(0°/90°)_*T*_/0°_*n*_/(0°/90°)_*T*_], where (0°/90°)_*T*_ denotes a fabric reinforcement. Alshahrani et al. [155] stated that the highest energy of unidirectional C/E laminates made by the compression molding process was absorbed by the sequence [(0°/90°/45°/−45°)_2_/0°/90°]_*S*_. When using C/E tubes, generally, it emerged that longitudinally oriented fibres have less influence on crashworthiness performance compared to hoop (90°)-oriented ones [124,130], due to the dominant folding failure mechanism. Junchuan et al. [156] compared hybrid (aluminium–G/P) unidirectional ([0°]_8_), cross-ply ([0°/90°]_4_, [0°/90°]_2*S*_), and quasi-isotropic ([0°/45°/90°/−45°]_*S*_, [0°/±45°/90°]_*S*_) laminates, which revealed that the 90° outer layers caused the inner layer of 0° to be inserted between aluminium folds more effectively and increased the energy performance. It is worth observing the influence of the plies’ orientation on the energy performance of tubes, which strongly depends on the material type (fibre and matrix) and their failure strain. For this reason, to the best of the authors’ knowledge, the lack of studies in the literature directly concerning the effect of different NFC ply angles on the energy absorption of tubes suggests the necessity of an in-depth investigation of such cases.

On the other hand, the effect of the filament winding natural fibre angle on the SEA of tubes has also been investigated in the literature. For K/P tubes, no significant differences were observed in SEA values with increasing winding angles from 0° to 5°. However, a decrease was noted for up to 10°, leading to trigger shear forces [62]. On the other hand, Supian et al. [32] observed an SEA value increase of almost double the original value when the winding angle was improved from 30° to 70° for the hybrid K-G reinforcement, since the wide axial profile of lower angles led to a decrease in axial stress on both the inner and outer surfaces of the tubes.

These studies suggest that properly choosing the composite stacking sequences and winding fibre angle could allow for an improvement in the structure’s crash performance properties. In particular, the stacking sequence design should aim to achieve a composite structure, which is able to absorb a high amount of energy during the impact and protect the inner structure from very large deformations. Generally, it emerged that the load–displacement response of laminates shows a greater energy absorption capability for cross-ply and quasi-isotropic stacking sequences compared to unidirectional ones [157]. Moreover, setting the outer plies’ fibre orientation to 90° can help to enhance the laminate crashworthiness performance, leading to an increase in P_max_ and P_avg_. Additionally, considering the importance of stacking sequence’s influence on crashworthiness, it is possible to directly optimise the composite design in terms of the number of plies, their material, and their thickness to maximise the absorbed energy during an impact event [158,159].

### 4.4. Strain Rate

The failure modes and the mechanisms involved in crushing are responsible for the energy absorption and are strain-rate-dependent. Hence, it is important to understand how the crushing speed influences the energy absorption ability of the composite components. In this sense, it is crucial to consider the mechanical properties of the matrix that are affected by strain rate. In particular, some resins, such as vinylester, polyester, and epoxy, were found to become increasingly brittle under elevated strain rates [35]. The higher fracture toughness of the composites allows for better control of the propagation of longitudinal cracks, and this allows for a higher AE to be obtained by making the fronds bend to the inner side and outer side, with a small curvature. On the other hand, using a matrix with specific mechanical properties might increase the stability of the part when facing local buckling caused by inertia effects during dynamic tests. Thus, higher magnitude forces would be required to cause the structure’s walls to collapse.

Generally, a quasi-static compression test is performed to obtain composites’ performance under impact at a low speed (according to ASTM D695-15, this is 1.3±0.3 mm /min, at least before the yield point [160]). Testing the specimen under quasi-static conditions allows for its fracture mechanisms to be identified and properly analyzed. However, to simulate the real-word behavior and resistance of a structure under crushing, it is fundamental to dynamically test the components.

Some of the literature focusing on fibre-reinforced epoxy tubes [148,161,162,163,164] reported that such composites exhibit a higher CFE and SEA under quasi-static tests compared to dynamic ones (speed of up to 10 m/s), with an increase of up to 40%. All of them reported that the energy absorption mechanisms were determined by the strain-rate-dependent crushing mechanism, with an increasingly brittle matrix occurring at high speeds. As explained later in Section 5, the differentiation of the dominant crushing mechanisms is crucial, and also can be used to understand the influence of speed on crashworthiness performance, as reported below. However, with respect to the mentioned works, Farley [130] observed, for C/E cylinders, that SEA positively varied with the speed of the test, ranging from 0.01 m/s to 12 m/s. This variation depended on the mechanism governing the crushing of the tube, according to the direction of the plies (θ). Interlaminar crack growth, which is primarily controlled by the matrix, drove the crushing of C/E tubes ([$\pm \theta_{3}$]); in this instance, an increase of 35% in energy absorption was recorded as the speed increased. Conversely, when the crushing process was characterized by the formation of lamina bundles, primarily influenced by fibre characteristics and therefore not dependent on strain rate, the SEA value of C/E tubes ([0°/$\pm \theta_{2}$]) was not influenced by crushing velocity.

López et al. [39] tested, both quasi-statically (speed of 60 mm/min) and dynamically (speed from 1.9 m/s to 9.1 m/s), tubular samples ($V_{5}$,$V_{6}$$V_{10}$,$V_{11}$) with a different kind of matrix: thermoplastic PLA (reinforced by flax and hemp-kenaf). In this case, the tested specimens absorbed more energy in dynamic tests than quasi-static ones (see Figure 11), whose SEA obtained values up to 40% less than those of their dynamic counterparts. Here, the greater ability to absorb energy was due to the inertia effects. In this regard, it is worth observing that PLA is a thermoplastic matrix, which is less stiff and strong compared to the thermoset matrices on which the other results were based.

In general, the effect of crushing speed on energy absorption ability is contingent upon factors such as material properties, orientation, and geometric configuration. As a result, it is challenging to define a universal rule.

## 5. Crush Load–Displacement History and Failure Modes

The design of a crashworthy component has to consider the failure modes during an impact event, which are responsible for the absorbed crushing energy and, hence, the crashworthiness. In this regard, it is crucial to avoid a sudden collapse of the crushed component, which corresponds to a large P_max_ value followed by a steep drop in the load–displacement curve. The combination of fibres and matrix, as well as the geometry of the tube, determine whether a tube will progressively crush. When interlaminar cracks are constrained, unstable cracks grow, or no buckle forms, a catastrophic failure occurs. The design phase of thin-wall structures should consider the necessity of obtaining a gradual failure, which, at the same time, makes them lightweight structures capable of absorbing a significant amount of energy. The load–displacement response of a crushing test serves as a valuable tool for detecting the various phases of failure and determining whether a catastrophic or progressive collapse occurs. It is particularly crucial to analyze and differentiate between the different failure modes exhibited by a composite material, specifying which modes are acceptable in terms of crashworthiness. By correlating the load–displacement response with the observed failure modes, a comprehensive understanding of the behavior of the material under crushing conditions can be achieved, aiding in the evaluation of its crashworthiness performance.

When designing a crashworthy component, it is essential to ensure that the absorbed energy increases while the transferred load gradually decreases through a progressive failure mode, which can limit high-impact loads. For this reason, load–displacement curves corresponding to some of the previously described tubular structures are considered from both geometrical and material perspectives.

First of all, it is essential to distinguish the possible failure modes characterising the composite under crushing. Brittle-fibre-reinforced composites typically exhibited transverse shearing (or fragmentation mode) and lamina bending (or splaying mode). Local buckling could also occur in ductile-fibre-reinforced composites. Moreover, brittle fracturing, as a combination of the transverse shearing and lamina bending, characterizes most of the brittle-fibre-reinforced composites [65]. For each of these modes, it is opportune to focus on the main mechanisms of the absorbing energy: the interlaminar crack growth and lamina bundle fracture for lamina bending. The former depends on the mechanical properties of fibres and their matrix and fibre orientations. Lamina bundles form from interlaminar and longitudinal cracks. They fracture when the stress on their tensile side exceeds the material strength. In the lamina bending mode, the lamina bundles do not break, but keep bending forward on the outside. Here, the matrix crack growth is the main factor responsible for absorbing energy, together with the frictional effects of lamina bundles with the loading surface. The crushing characteristics of this mechanism are shown in Figure 12b.

A wedge-shaped laminate cross-section with one or more short interlaminar and longitudinal cracks characterizes the transverse shearing crushing mode, as shown in Figure 12a. Those cracks form partial lamina bundles and their fracture is the main source of energy absorption, together with the growth of interlaminar cracks. The local buckling crushing mode verifies when the strength of the matrix exceeds the interlaminar stress in brittle-fibre-reinforced composites, and its failure strain is superior to that of fibre such that it exhibits plastic deformation even under high stress. Through the plastic deformation of the material, local buckles form, as shown in Figure 12c. For ductile-fibre-reinforced composites, integrity is also maintained after crushing, since the fibres and matrix can withstand high levels of stress without fracturing. A combination of the last two failure mechanisms leads to brittle fracturing (Figure 12d). Here, lamina bundles can fracture. An energy absorber should exhibit transverse shearing, lamina bending, or local buckling crushing modes to obtain the desired decelerating history during a crash so that vehicle safety is ensured.

From a damage perspective, NFCs differ from synthetic-fibre-reinforced composites due to the unique characteristics of natural fibres, as revealed in stress–strain curve analyses. While NFCs typically exhibit non-linear behavior, synthetic ones tend to display linear behavior until failure, as shown in Figure 13.

The inherent elasto-plastic behavior of natural fibres enables them to deform more than their synthetic counterparts without breaking, enhancing their effectiveness in dampening vibrations and providing advantages in applications such as automotive or aerospace crashworthiness components [166,167]. On the other hand, natural fibres typically have more heterogeneous micro-structures, which can lead to micro-structural damage occurring due to the breaking of individual fibres, delamination between layers, and the separation of fibre bundles [168]. Micro-fibrils characterize cellulose-based fibres and govern their stiffness. In particular, the micro-fibril angle contributes to the mechanical performance of NFCs [168]. When the micro-fibril angle aligns with the loading direction, NFCs exhibit enhanced elastic properties. In [169,170,171], it is reported that the larger the angle, the higher the failure strain, as the fibrils can twist during stretching.

Yan et al. [25], focusing on the quasi-static compression, observed that most specimens with N = 2, 3, regardless of the D and L/D, exhibited progressive failure, where the cracks in the top or bottom of the tube propagate, leading to the formation of inward and outward fronds. However, the one-layer specimens typically failed, with irregular deformations. The successive study by the same authors [24] reported the effect of triggering F/E circular tubes, whose load–displacement curve is shown in Figure 14. No significant differences in the failure modes were observed after the triggering. The observed dominant failure mechanisms were the longitudinal crack growth and the formation and fracture of lamina bundles. The triggering mechanism significantly influenced the origin and magnitude of the peak load, as well as enhancing the stability of the load–displacement response, indicating a greater CFE.

It is evident that a trigger mechanism does not affect the stiffness of the structure, since the slope of the load–displacement response before the first peak load is unchanged. Some of the specimens (non-triggered and those with the smallest N) of this experimental study presented an initial catastrophic failure due to a circumferential crack starting at the middle of the tube, followed by a stable crushing behavior. In this case, after the first peak load, the curve drop was large and there were high fluctuations in the average load value. Ochelsky et al. [46] analyzed the crashworthiness performance of C/E and G/E circular tubes. In both cases, the specimens were filled with PVC or aluminium (Al) foam. In addition to the effect of filling, the load–displacement curves led to a significantly higher P_avg_ value with respect to the empty case, very few fluctuations around it, and a small drop after the first peak load. Hence, these two configurations led to progressive failure with a good energy absorption capability. All the tested specimens were crushed through a layers bending mode (only on the outside of the tube in case of filling) and defragmentation.

For hybrid composites, it should be considered that the fibre features of the outer layers typically determine the crushing mode. Strohrmann et al. [162] tested cylinders with four different hybrid configurations, varying the stacking sequence as follows: [C]_8_-, [C_2_/F_2_]_*S*_ (C-F)-, [F_2_/C_2_]_*S*_ (F-C)-, and [F]_8_-reinforced cylindrical tubes. The structure reinforced solely with carbon fibre exceeded the other configurations in terms of crashworthiness performance. On the other hand, C-F and F-C tubes slightly differed in terms of their energy dissipation and average force (higher in the first case). The deeper oscillations around the mean load value of C-F tubes resulted from round components, which radially cut the cylinder before tiny pieces were created to surround the tube (fragmentation mode). Using this study, it is possible to compare the carbon- and flax-fibre-reinforced composites’ failure mechanisms. There is a difference in crushing mode due to the brittle nature of the former in contrast with the ductility of the flax composite [162]. Hybrid tubes with external carbon or purely carbon-fibre-reinforced tubes crushed in fragmentation mode, while pure flax or the use of the outer layer led to a splaying failure mechanism. The results demonstrated that the use of flax fibres in the outer layers was able to enhance the impact resistance, potentially impeding linear crack propagation. Moreover, contingent upon the arrangement of layers in the composite structure, the carbon–flax composite may exhibit a more brittle or a more ductile response. When subjected to tensile forces, the hybrid composite tended towards the brittleness of carbon if it formed the outer layer, while it tended towards the ductility of flax if it occupied that position. Nonetheless, specific fibre fractions and layer arrangements can have a mixed influence, enhancing one property while compromising another. For instance, incorporating flax properties into a non-hybrid carbon composite bolstered damping and impacted resistance but resulted in diminished tensile and bending strength. Another instance of this is the advantageous placement of flax fibres on the outer layers in hybrid composites, which leads to improved impact resistance, while outer layers composed of carbon fibres confer greater stiffness.

Meredith et al. [60] explored the potential of natural fibre cones as energy absorbers, paying particular attention to the performances of woven flax and jute and unwoven hemp. The hemp cones showed progressive failure if subjected to dynamic axial compression, mainly exhibiting lamina bending and reduced brittle fractures. On the other hand, the jute and flax failed due to brittle fracturing after an initial lamina bending. The greater the fibre density, the higher the resistance to longitudinal crack propagation for dynamically tested F/E (average V_f_ = 37%), J/E (average V_f_ = 31%), and H/E (average V_f_ = 47%) cones. Significant variability in crashworthiness performance also resulted from the fibre strength variations and the manufacturing process. Longitudinal cracking in resin-rich areas was a source of energy dissipation, while insufficient attenuation prevented energy from dissipating through lamina bending. The best composite cone tested in this study was reinforced by hemp, with the highest recorded fibre volume fraction (V_f_ = 48%).

Ataollahi et al. [172] reported experimental results on the failure response of square S/E tubes. When subjected to axial crushing, the tested specimens showed a ductile deformation without debris splitting after crushing, in contrast with brittle-fibre-reinforced composites (e.g., C/E or G/E). However, the significant drop after the peak load suggested a catastrophic failure (Figure 15).

Even though energy can be absorbed by structural elements experiencing catastrophic failure, it cannot be considered a suitable mechanism for preventing the deceleration of an object during a crash. Additionally, as already observed, the poorer mechanical properties of silk fibre and their superior cost with respect to the conventional natural reinforcement used in laminates means that it is very rarely used in practical applications.

## 6. Outlook and Challenges

In recent years, NFCs have attracted considerable attention across several industrial sectors and research fields owing to their notable benefits in terms of their sustainability, lightweight construction, and cost-effectiveness. While natural fibres boast unique properties such as a high specific strength, low density, and efficient energy absorption, making them well-suited to specific energy absorption applications, they still face a performance gap compared to high-performing synthetic counterparts, such as carbon fibre, as discussed earlier. To address this gap, recent research efforts have shifted focus towards strategies aimed at overcoming the existing challenges associated with the exclusive use of natural fibres. Moisture sensitivity emerges as a critical factor influencing the mechanical properties of composite materials, especially in natural environments where moisture absorption can lead to significant degradation [173,174]. The hydrophilic nature of natural fibres poses challenges for outdoor applications and compromises their durability. Several parameters affect the water absorption of composite materials, including the choice of polymers and plant fibres, the fibre mass fraction, the modification techniques, the absorption duration, and the environmental humidity. Strategies such as hybridising plant fibres with synthetics, applying polymer coatings, and employing tile process coatings have shown promise in enhancing the composite material’s water resistance. Additionally, the incorporation of nanofillers has proven effective in mitigating water absorption and improving mechanical properties in humid conditions [175]. Moreover, chemical methods, including alkali treatment, benzoylation, and acetylation, have also been explored to reduce the hydrophilicity of plant fibres, thereby enhancing the composite material’s durability in moisture-rich environments, as well as enhancing the mechanical properties [69,176]. Actually, exploring various chemical and physical methods that could improve the matrix–fibre interfacial bonding and surface properties is crucial. Moreover, fibre consistency, fibre quality, and diameter variations, along with a poor fibre–matrix interface, affect the composite performance. In this sense, the optimisation of the manufacturing process could help to enhance the final mechanical properties by improving the quality of the fibre–matrix interface and reducing defects [177,178,179].

Furthermore, cultivating natural fibres in region-specific areas where they thrive naturally reduces the need for excessive irrigation, fertilizers, and chemical treatments, leading to a more sustainable agricultural practice. This can bolster sustainability and mitigate the environmental impact, aligning with worldwide endeavors to address climate change [180].

To tackle these challenges effectively, significant research efforts are required to ensure the reliability and optimisation of natural fibre composites’ performance in various applications. Finally, the use of computational and numerical modeling techniques is crucial for predicting energy absorption properties without experimental measurements. However, accurate input data availability remains a challenge. To the authors’ knowledge, only a few works have focused on developing reliable modeling for NFC tubular structures [181,182].

## 7. Conclusions

The present study conducted a comprehensive investigation into the crashworthiness of NFCs and hybrid tubular structures during axial crushing. Its main goal was to highlight the potential of these composites, which incorporate natural fibres, as structural energy absorber components. The influence of both geometrical and material factors on the energy absorption capabilities of thin-walled structures is discussed. In particular, circular cross-sections emerge as the preferred design for tubular structures, demonstrating superior energy absorption compared to their square counterparts. While NFC tubes can achieve energy absorption levels similar to GFRPs, they fall short of those attained by CFRPs. Hybridisation emerges as a promising strategy to mitigate the performance reductions associated with the inclusion of natural fibres.

The optimisation of energy absorption performance while minimising high peak loads is crucial for enhancing crashworthiness during impact events. Notably, trigger mechanisms play a pivotal role in promoting progressive failure and reducing peak load values, thereby enhancing occupant safety and structural integrity in automotive applications. Additionally, foam filling can expand the plastic deformation zones and enhance the energy absorption efficiency of the tubes, potentially reducing the maximum load values. In contrast to tubes, conical structures do not require crush initiators, with the vertex angle playing a critical role in crashworthiness performance. Lower cone inclinations generally yield higher energy absorption values.

Moreover, the strain rate affects the energy absorption performance of composite components. Since its influence on crashworthiness depends on the material type and dominant crushing mechanisms, establishing a general rule regarding the differences in performance when transitioning from quasi-static loading conditions to dynamic ones proves challenging.

Overall, despite the environmental benefits of NFCs, they are not yet viable replacements for synthetic-fibre-reinforced composites in structural applications, such as automotive crash-boxes. Addressing this limitation requires delving into future research challenges, and may involve exploring potential solutions such as matrix modifications, the chemical treatment of fibres, reinforcement with fillers, and optimisation of the manufacturing process. By delving into these areas, researchers can work to enhance the competitiveness of NFCs and bridge the gap between them and their synthetic counterparts, ultimately paving the way for broader applications and greater sustainability in various industries.

Finally, it is imperative for the design of energy absorber structures to harness the predictive capabilities offered by accurate numerical modeling. Through numerical simulations, designers can obtain a deeper insight into the behavior of tubular composites under crushing conditions. This invaluable insight could streamline the design process by allowing for the identification of optimal structural configurations and material compositions. Leveraging numerical simulations can lead to cost savings by minimising the need for expensive testing of this prototype and further iterations. Ultimately, the integration of accurate numerical modeling into the design process has the potential to enhance the development of energy absorber structures, making them more efficient, reliable, and cost-effective.

## Figures and Tables

**Figure 2 materials-17-02246-f002:**
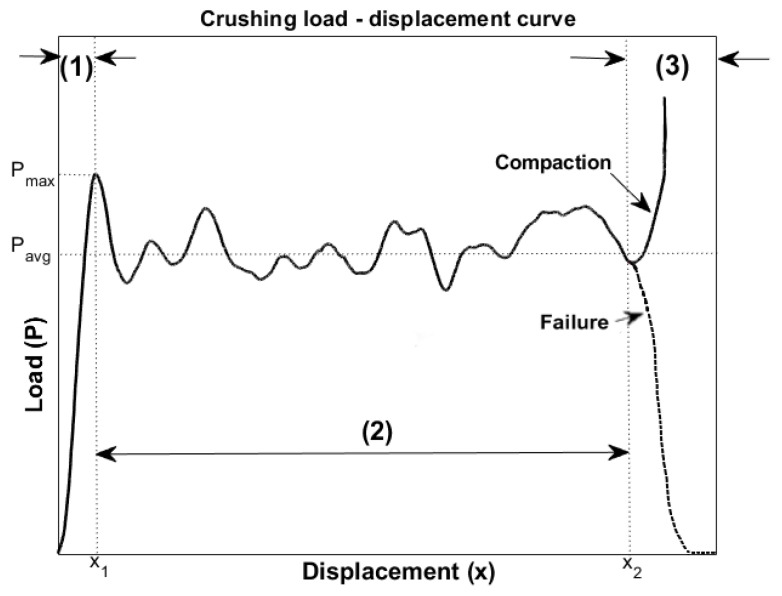
Typical load–displacement curve of composite subject to axial crushing: (1) pre-crushing zone; (2) post-crushing zone; (3) compaction or failure zone.

**Figure 3 materials-17-02246-f003:**
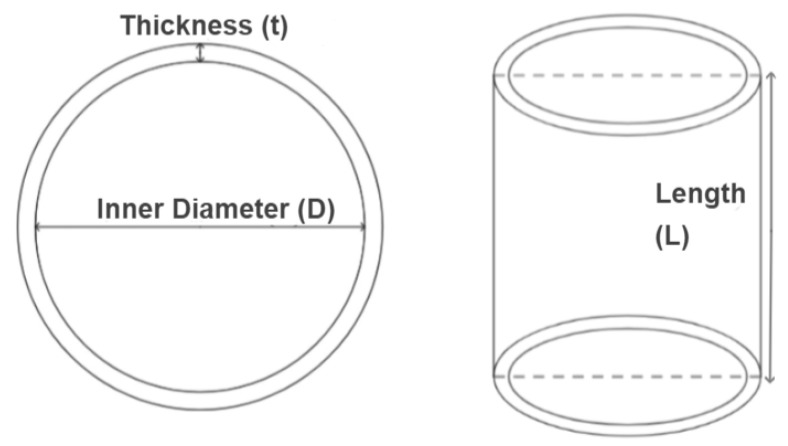
Circular cross-sectional thin-walled tube.

**Figure 4 materials-17-02246-f004:**
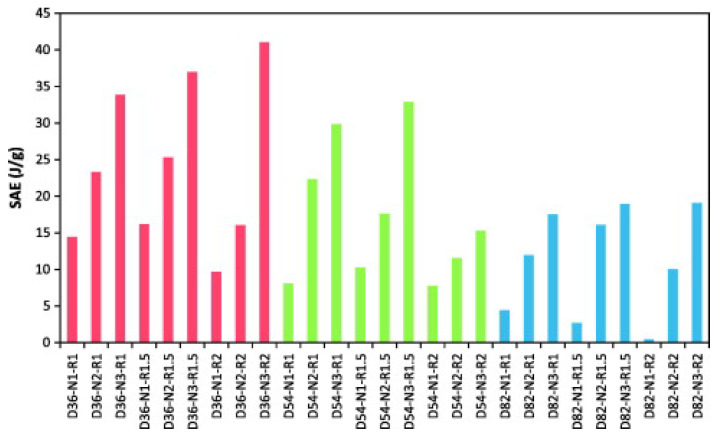
Comparison of SEA values of circular F/E specimens tested under quasi-static axial crushing by Yan et al. [25]. D: internal diameter, i.e., 36 mm (**red**), 54 mm (**green**), 82 mm (**blue**); N: number of layers; R: length-to-diameter ratio.

**Figure 5 materials-17-02246-f005:**
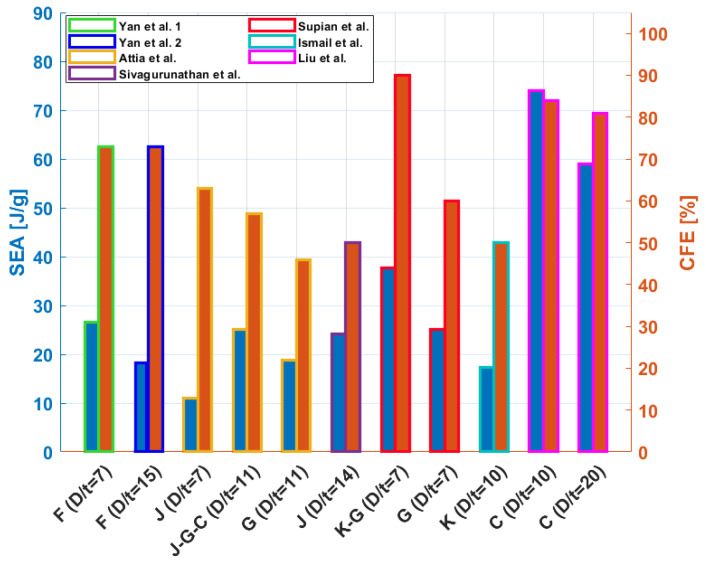
SEA (blue) and CFE (orange) of circular tubes, according to the reinforcement type and D/t ratio, obtained by Yan et al. (1) [25], (2) [26], Sivagurunathan et al. [28], Attia et al. [29], Liu et al. [31], Ismail et al. [62] and Supian et al. [32]. Flax (F), jute (J), glass (G), kenaf (K), and carbon (C).

**Figure 6 materials-17-02246-f006:**
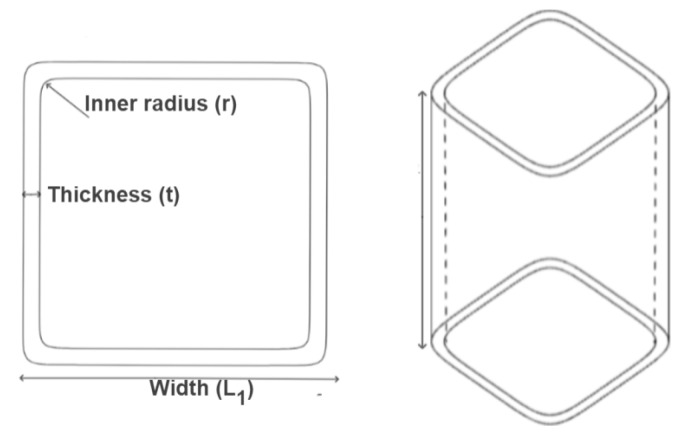
Square cross-sectional thin-walled tube.

**Figure 7 materials-17-02246-f007:**
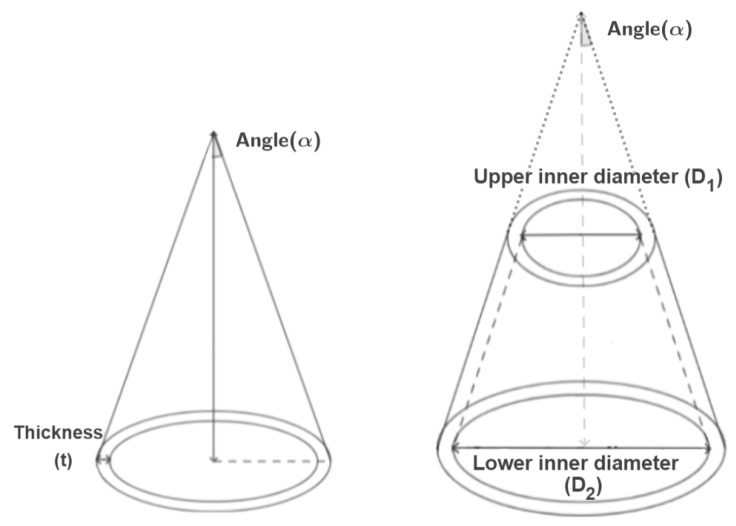
Conical and truncated conical thin-walled tubes.

**Figure 8 materials-17-02246-f008:**
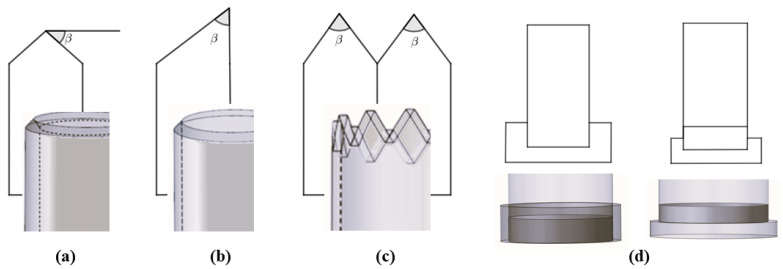
Frontal view of some of the most used triggers for tubes: (**a**) single external chamfer (bevel); (**b**) double chamfer (steeple); (**c**) tulip; (**d**) plug-types.

**Figure 9 materials-17-02246-f009:**
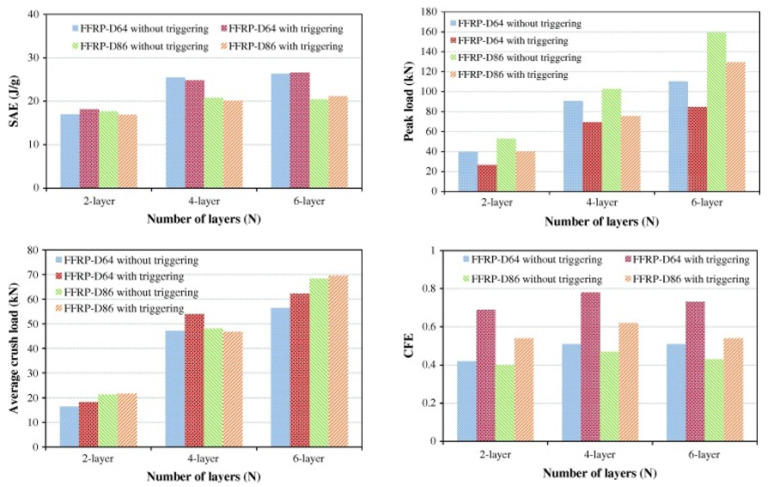
SEA, P_avg_, P_max_, and CFE comparison of 45°-chamfered and non-triggered cylindrical tubes subjected to quasi-compression tests [24].

**Figure 10 materials-17-02246-f010:**
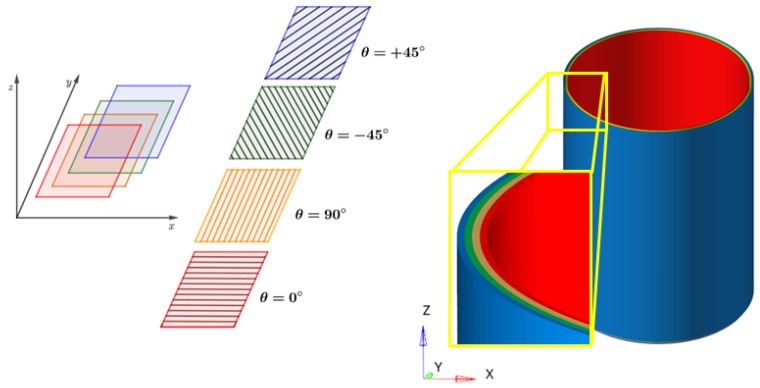
Outline of flat and tubular laminated structures made of stacked plies with different fibre orientations.

**Figure 11 materials-17-02246-f011:**
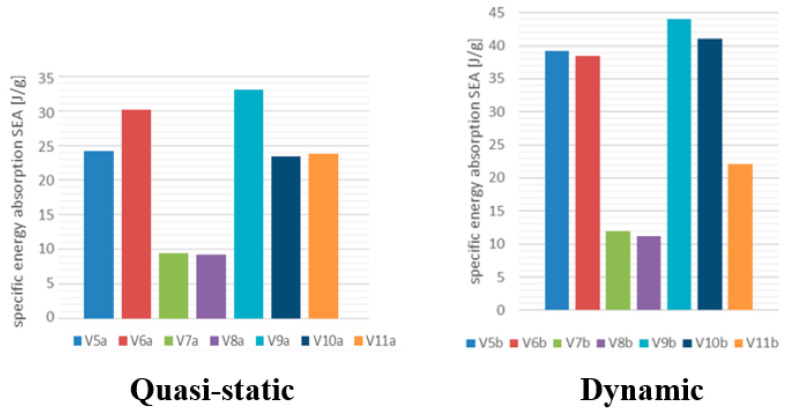
Comparison of the specific energy absorption of NFCs (with PLA or HDPE matrix) tubes under dynamic and quasi-static loading conditions [39].

**Figure 12 materials-17-02246-f012:**
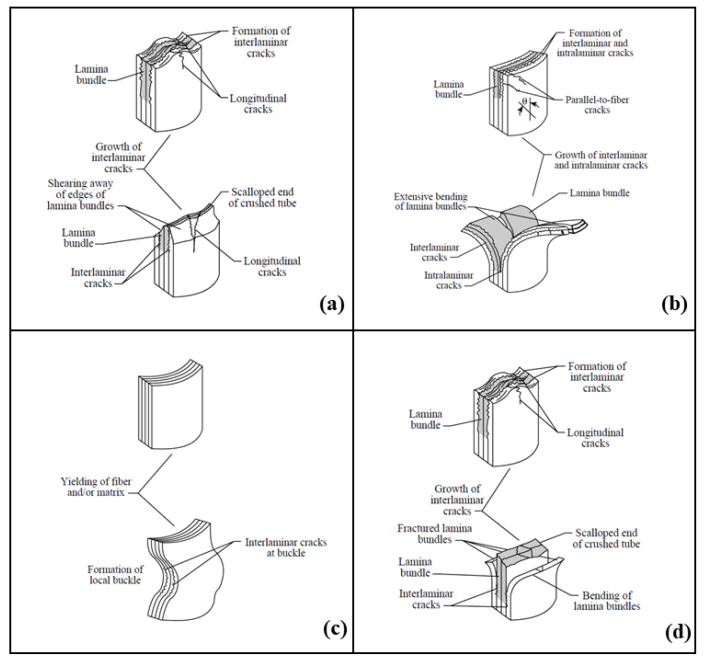
(**a**) Transverse shearing; (**b**) lamina bending; (**c**) local buckling; (**d**) brittle fracturing crushing modes [65].

**Figure 13 materials-17-02246-f013:**
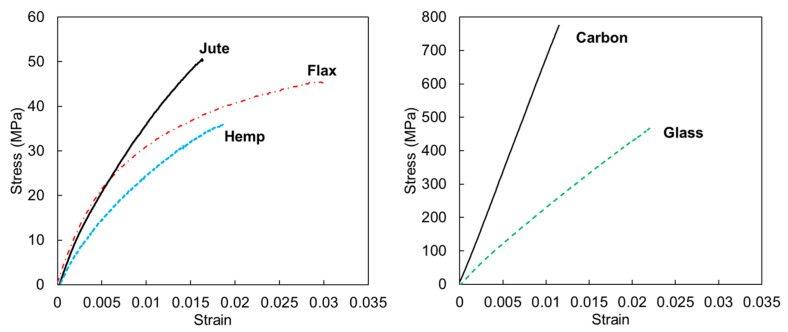
Exemplary stress–strain curves of natural fibre composites (**left**) and synthetic ones (**right**) [165].

**Figure 14 materials-17-02246-f014:**
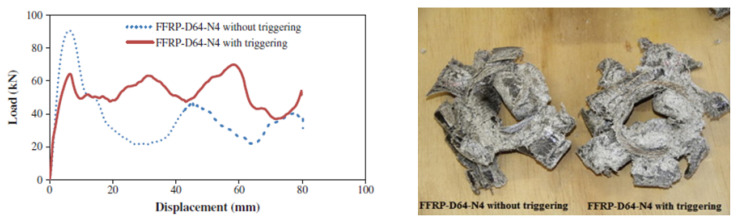
Load–displacement response (**left**) and crushed F/E triggered and non-triggered cylinders subjected to quasi-static compression (**right**) [24].

**Figure 15 materials-17-02246-f015:**
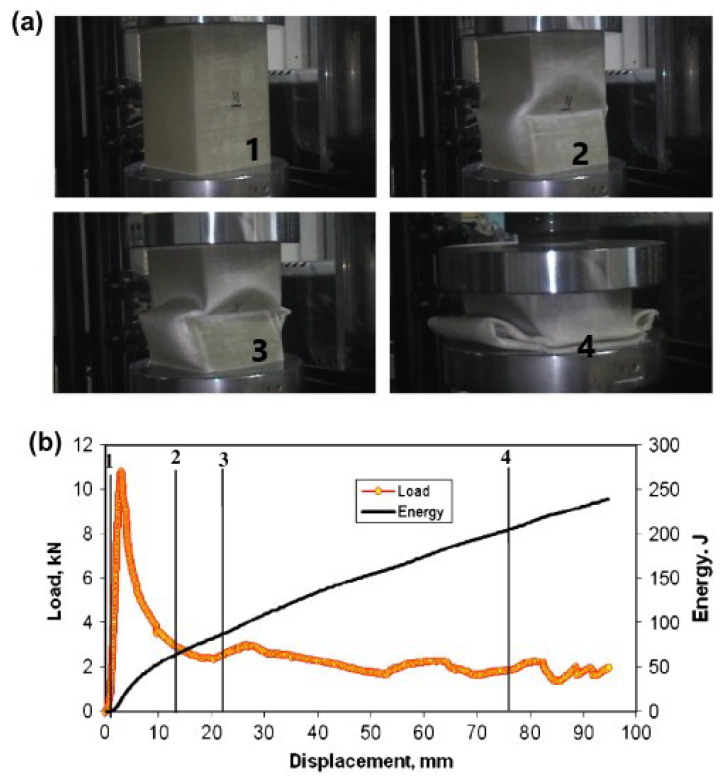
(**a**) Crushing history and (**b**) load/energy-displacement curves of a S/E square tube subject to quasi-static crushing [172]. The sequential number 1–4 marks the point of the two curves corresponding to the photographs.

**Table 2 materials-17-02246-t002:** Material and mechanical properties of HDPE and PLA matrices (reinforced with flax, hemp and kenaf in cylinders subjected to axial crushing) [39].

Matrix	Density [g/cm^3^]	Young’s Modulus [GPa]	Tensile Strength [MPa]	Elongation at Break [%]
HDPE	0.95–0.97	0.55–1.1	20–37	10–1200
PLA	1.21	3.3	30	2.5

**Table 3 materials-17-02246-t003:** Comparison of SEA and CFE of the overviewed square fibre-reinforced epoxy tubes.

Material	L_1_/t	L [mm]	SEA [J/g]	CFE	Reference
Ct/E	81	100	2.1	∼0.5	[53]
J/E	13	100	22.3–31.3	0.6–0.8	[51]
K/E	8	350	22.4	0.46	[52]
R/E	47	50	4.2–4.8	0.2	[50]
S/E	47	50–120	4.0–5.3	0.4	[48,49]
C/E	28	100	77.0	∼0.7	[54]
C/E	12	100	74.2	∼0.7	[36]

**Table 4 materials-17-02246-t004:** Comparison of SEA and CFE of the studied conical natural-, glass- and carbon-fibre-reinforced tubes.

Material	*α*	h [mm]	SEA [J/g]	Reference
Cc/P	0°, 12°, 24°, 36°, 48°, 60°	100	0.1–0.6	[59]
Cc/P	5°, 10°, 20°	110	∼2.0–9.0	[62]
Ct/E	5°, 10°, 20°	110	∼6.0–12.0	[61]
F-G/P	30°	110	∼20.4	[61]
OP/E	0°, 6°, 12°, 18°	100	3.4–6.7	[58]
OP/P	0°, 12°, 24°, 36°, 48°, 60°	100	0.4–0.6	[59]
C/E	0°, 6°, 12°, 18°	100	23.0–29.0	[58]
C/E	0°, 5°, 10°, 15°	110	36.4–87.4	[63]
G/E	0°, 5°, 10°, 15°	110	32.6–77.4	[63]

**Table 5 materials-17-02246-t005:** Comparison of SEA values for circular F/E, Bb/E, G/E, G/P, and C/E empty or foam-filled tubes [24]. Al: aluminium; PMA: polymethyl acylamid; PU: polyurethane; PVC: polyvinyl chloride.

Material	Trigger	Foam	SEA [J/g]	Reference
F/E	No	No	17.0–26.3	[24]
	No	PU	18.4–28.7	[24]
	45° bevel	PU	18.1–28.8	[24]
Bb/E	No	No	1.2–4.0	[47]
	No	PU	0.7–4.6	[47]
G/E	No	PU	14.5	[45]
	No	PMA	9.1	[45]
C/E	No	No	46.3–48.3	[46]
	No	PVC	35.2–46.5	[46]
	No	Al	28.2–40.3	[46]
G/P	No	PU	17.0	[44]
	45° bevel	PU	14.8–19.7	[33]

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
