# Peer review of "Natural Fibre and Hybrid Composite Thin-Walled Structures for Automotive Crashworthiness: A Review"

_materials, 2024, doi:10.3390/ma17102246_

Round 1
Reviewer 1 Report
Comments and Suggestions for Authors
The authors did a really good job with the scientific content of this review article. A lot of important aspects of the fiber reinforced composites are discussed in this study. They did an extensive literature review and discussed the major key points successfully. I have some minor edits and corrections listed below.
Also, I would strongly advice the authors to do some moderate English editing before re-submitting. There are several grammar issues throughout the entirety of the manuscript. Most of them do not critically affect the readability of the document, but some of them do. I tried to correct as many as I could, but a once over from a native speaker, or maybe the English/writing department of the University would be beneficial.
Line 2: Please remove comma after the word absorption “…energy absorption are increasingly emerging…”.
Line 4: Please remove comma after the word applications “…applications due to inferior…”.
Line 18: Please use semicolon and not comma after the word “hybrids” like so “hybrids;”.
Line 34: The use of “over the period of…” is typical when we sum the years (e.g., over the period of 12 years). Please change this sentence as follows: “…in road accident fatalities between 2012 and 2022”.
Line 39: Please change “pillars” to “pillar”.
Line 51: Please remove the comma after the word “structures”.
Line 58: Please change the second part of the sentence as follows: “…designers of energy-absorbing structures must possess a thorough understanding of the structure-property relationship in these composite systems”.
Line 63: Please start the sentence as follows: “This study conducts a detailed…” the phrase “in particular” is not necessary here.
Line 77: Please replace “as exists for” with “as compared to”.
Line 78: Please remove the word “primarily”.
Line 83: I believe the sentence here referring to Section 3 is actually referring to Section 4 of the manuscript. The previous sentence starting in line 82 seems to be referring to Section 3 (crashworthiness). If so, please correct the sentences accordingly.
Line 101: Please add the work “the” and delete the comma as follows: “…depending on the mechanical properties of the materials and their architecture”.
Line 102: Please delete this sentence “A significant difference between….”, is not need here.
Line 107: Please reword the end of the sentence as follows: “,composites are increasingly replacing metals, despite their brittleness comparted to the ductility of the latter”.
Line 110: Please remove the comma after the word composites.
Line 112: I do not recommend starting a new paragraph with a contradiction following the previous paragraph. A new paragraph introduces a new concept/ides, so you can start directly with the word “nowadays”, as such “Nowadays, the drive for…”.
Line 130-131: Please add citations to this sentence. Also, besides they high energy consumption and high cost of the mechanical and thermal processes for recycling composites, I would add the harsh and non-environmentally friendly chemicals/solvents used for chemical recycling.
Line 150: Please change the sentence as follows: “natural ones are biodegradable and come from renewable resources”.
Line 159: Please change as follow: “Furthermore, the increasing emphasis on sustainability…”. You need an adjective here “increasing”, not an adverb “increasingly”.
Line 169: Please re-write the sentence as follows and connect it with the next paragraph. As it stands, it is a one sentence paragraph: “Natural fibre composites have already found application in…trunk liners, and more. However, the utilization of purely NFCs in structural…”.
Lines 189-191: This sentence needs citations.
Line 193: Please the word “crushing” to “crush”.
Line 199: please replace the word “stage” with “phase”.
Figure 2: Does the figure present data collected by the authors? If not, please add citation. If it is a graph taken from another publication and modified by the authors, please state that, and add citation.
Lines 206-207: This sentence needs to be re-written. Suggestion: “The maximum load recorded during impact (Pmax), excluding the post-crushing zone and the potential increase of the load due to compaction”.
On the next bullet point, following line 207: Please use comma after “i.e.,” or put the abbreviation “AE” in parenthesis “The total absorbed crushing energy (AE), which is the amount…”. Please correct that in the following lines/sentences, or where necessary.
Line 215: Please change the sentence as follows: “…the higher the SEA, the greater the performance of the component will be as an energy absorber”.
Line 217: Please change the word “related” to “proportional”.
Line 220: Please change the phrase “On the other hand,” with “On the contrary,”.
Line 222: I believe you mean “passenger vehicle” and not “vehicle passenger”, please correct.
Line 223: Please re-write the sentence as follows: “Consequently, achieving a CFE value close to one, indicates minimal deviation of Pavg from the peak. This could help mitigate the deceleration perceived by passengers.”. Also, this whole section and the equations presented need citations.
Line 242: Please change the sentence ass follows “…configurations, comparing them to some examples of common synthetic fibre-based composites.”.
Line 276: Please remove “to” before “quasi-static” (“…tested under quasi-static conditions, to…”).
Line 279: The way the sentence is written, it is not clear what is the value for the ratio R. Did that sample have the lowest or highest value of L/D? Please re-write this sentence and clarify.
Lines 291-298: A new paragraph should start with this sentence “In a later study, authors investigated glass/epoxy (G/E)…using flax”. Also, the two sentences in lines 294 and 294 should be in the same paragraph and merge them with the paragraph that follow in line 298. These should all be in the same paragraph.
Line 321: Please connect this paragraph with the previous sentence, since they are on the same subject – “…a sudden collapse occurred. From the analyzed results, flax fiber in…”. It is not recommended to have once sentence paragraphs.
Line 333: Please add units next to the number 27, as follows “(27 J/g vs. 25 J/g)”.
Line 378: The abbreviation for high density polyethylene is HDPE, please replace “HD-PE” in the text with “HDPE”.
Line 381: Please replace the word “either” with “both” as follows “Indeed, both in static and dynamic tests, HDPE based…).
Line 396: Please fix the margins in this paragraph.
Line 403: Should be “lengths” not “length”, also add units after every number in the parenthesis “(50 mm, 80 mm, and 120 mm)”.
Line 416: add units after the value 100 and reverse the numbers as follows, should be “…from 50 mm to 100 mm while…”. Please make sure every number value in the text has the unit right next to it (e.g., line 428 units after 74, etc.).
In table 5, please correct the word “polymethylacylamid” to “polymethyl acrylamide” also the abbreviation for this family of polymers is not “Pl”, but “PMA”.
Figure 11, the axis labels are hard to read. Please re-write the letters and numbers in the figure with a different fond, or better resolution (you can use software to do that, such as Illustrator, Photoshop, PowerPoint, etc.).
Please make sure you are consistent with the use of the word “fibre”. In some instances, the word “fiber” is used instead. Please replace where necessary.
Comments on the Quality of English LanguageI would strongly advice the authors to do some moderate English editing before re-submitting. There are several grammar issues throughout the entirety of the manuscript. Most of them do not critically affect the readability of the document, but some of them do. I tried to correct as many as I could, but a once over from a native speaker, or maybe the English/writing department of the University would be beneficial.
Reviewer 2 Report
Comments and Suggestions for Authors
This review cannot be considered as completed unless the mechanisms of degradation are not presented.
Authors should add the explanation of the processes which caused degradation from structural point of view.
Comments on the Quality of English LanguageEnglish is fine.
Reviewer 3 Report
Comments and Suggestions for Authors
This review provides a comprehensive overview of the energy absorption characteristics of NFCs and synthetically composite materials under axial compression conditions. However, there are the following suggestions for modifications in the analysis of experimental results and the structure of the paper's discussion:
1、The beginning of Section 2 introduces NFCs through the narrative of “the superiority of composite materials in the automotive industry” - “the importance of the circular economy” - “the challenges of synthetically composite materials in sustainable recycling and utilization,” but it is too descriptive and lengthy. Please add details to it and shorten it by focusing on the main point.
2、The last part of Section 3 listed the main factors affecting the durability of energy absorbed structures, but suggested that the author supplement relevant reference literature or provide explanations, such as “The research results of A indicate that B plays an important role in this regard.”
3、In Section 4.2 of the article, the author mentioned that adding a foam filler can significantly enhance the energy absorption efficiency of the structure, but did not further provide detailed quantitative analysis results. To ensure the rigor of the article, it is recommended to further supplement relevant experimental data and analysis results.
4、In Section 4.3, the author introduces the mechanical properties and failure modes of laminated plates. However, laminated plates are flat structures, while the subject of this review article is tubular structures. What is the purpose of the introduction to laminated plates here? If it is to further explain the manufacturing method of energy absorbed structures, it is recommended to draw relevant diagrams for supplementary illustration.
5、In Section 4.4, the author's research focuses on the influence of crushing speed, but the author did not clearly specify the speed range at the beginning. On the other hand, at the end of Section 4.4, the author wrote, "the effect of crushing speed on energy absorption capability is contingent upon factors," while in Section 6, it is stated that "The overview explores ... as well as the crushing speed of these components, all of which play crucial roles in their energy absorption capabilities and associated failure modes during crushing." Are these descriptions contradictory? Additionally, would it be more appropriate to change the title of Section 4.4 to "Strain Rate" for subsequent analysis?
6、The title of the article is "Natural Fibre and Hybrid Composite Thin-Walled Energy
Absorber Structures subject to Axial Crushing: A Review" However, the main focus of the study is on a specific type of automotive crash structure. It is recommended to modify the title accordingly to narrow down the scope.
Comments on the Quality of English Languagenone
Reviewer 4 Report
Comments and Suggestions for Authors
The paper gives a review on natural fibre and hybrid composite thin-walled energy absorbing structure subjected to axial loading. The grammatical status of the paper is good, however, many technical aspects in a review of this kind is lacking. Some of these aspects are listed for the authors to adequately addressed before their review manuscript can be considered in this journal.
In section 3, apart from the three zones listed, the authors should distinctively mention and explain the three stages in crashing – the elastic stage, plateau stage and densification stage.
The initial peak load should be mentioned and why it is very important. How is this load different from the maximum peak load?
Page 6, line 208, What is P in the equation? Please, assign numbers to all the equations.
Comparison of the different fabrication techniques used for producing energy absorbing composite structures should be listed and explained. Their strength and weakness as well as the influence of these techniques on crushing performance should be discussed. Are there state of the arts ways in obtaining NFC for improved crashworthiness application?
General deformation mechanisms involved in the energy absorbing composite structure should be elaborated. Is there deformation mechanism for NFC?
It is necessary to compare the crushing performance of various NFC energy absorbing structures with their counterparts. Propose possible ways to improve crushing performance.
There should be more discussion on the renewability, biodegradability and sustainability of energy absorbing NFC structures with environmental impact.
Some useful references are missing in the review paper which buttresses on green composites, polymer-based, fabrication and material-structural influences of crushing performance of energy absorbing structures. https://doi.org/10.1016/j.susmat.2020.e00196, https://doi.org/10.1080/17452759.2023.2273296, https://doi.org/10.1080/17452759.2023.2197436
A section elaborating the challenges, prospects and future trends must be included in the paper.
The authors should avoid using the word article for a paper not yet accepted. The conclusion should be revisited and rewritten.
Round 2
Reviewer 4 Report
Comments and Suggestions for Authors
Authors addressed most issues, their paper is accepted in this journal.